# From <Answer> to <Think>: Multidimensional Supervision of Reasoning Process for LLM Optimization

## Abstract

Large language models (LLMs) can develop strong reasoning ability when trained appropriately. Existing approaches are broadly categorized into outcome-level answer supervision and process-level reasoning supervision. However, the former provides only sparse binary feedback and overlooks intermediate step quality, while the latter scores individual steps but requires task-specific segmentation. To this end, we propose a novel framework that assesses the quality of reasoning process along three dimensions: **Confidence** for uncertainty calibration, **Relevance** for semantic alignment and **Coherence** for logical consistency. Together, these dimensions capture aspects beyond final answer correctness and enable interpretable assessment without requiring ground truth answers. Our framework serves as a **D**imension-level **R**eward **M**odel (**DRM**) that assigns scores to reasoning processes and provides supervision signals for both off-policy (e.g., DPO) and on-policy (e.g., GRPO) optimization. Experimental results show that DRM provides effective supervision signals, guides the optimization of LLMs and enhances their reasoning ability. In particular, DRM-supervised training achieves consistent gains on both in-distribution and out-of-distribution open-domain tasks, including mathematics, question answering, code execution and puzzles. Our findings demonstrate that multidimensional supervision of reasoning process can improve the generalized reasoning ability of LLMs beyond the training distribution.

## 1 Introduction

Enhancing the reasoning ability of Large Language Models (LLMs) to perform complex and multi-step reasoning remains a central challenge in their development (Zhang et al., 2025b; Xu et al., 2025). The dominant paradigm for enhancement relies on Reinforcement Learning with Verifiable Rewards (RLVR) (Shao et al., 2024; Yang et al., 2024; Luo et al., 2024). RLVR provides supervision at the outcome level, assigning a positive reward only if the final answer is correct. However, this reward mechanism has fundamental limitations. First, answer supervision overlooks the quality of the reasoning process (Yu et al., 2025a). This often leads to rewarding models for arriving at a *correct answer with flawed reasoning* while penalizing sound logic that contains a minor final error (Xie et al., 2025). Second, we observed that rewards in RLVR can become nearly constant when the model is either too powerful or too weak on the training set, thereby offering limited guidance for optimization (Cui et al., 2025). Process-level Reward Models (PRMs) are designed to address these limitations by supervising intermediate steps (Cheng et al., 2025; Zhang et al., 2025a; Zou et al., 2025). While promising, PRMs introduce their own challenges. Their process-level supervision requires the reasoning process to be segmented into individual steps (Xiong et al., 2025; Zou et al., 2025). This segmentation is often learned in a task-specific manner, which may hinder generalization to open-domain tasks with ambiguous or overlapping steps (Xiong et al., 2025). Furthermore, unlike the transparent binary signal of RLVR, PRMs often function as black-box evaluators, making it difficult to diagnose or trust their scoring mechanism (Christiano et al., 2023).

To overcome these limitations, we propose a new supervision framework grounded in the key characteristics of a high-quality reasoning process. Prior work shows that unfaithful content in reasoning process can hinder correct answers (Zhang et al., 2025b). To detect such content, our framework performs assessment along three complementary dimensions: (1) **Confidence**, measures whether

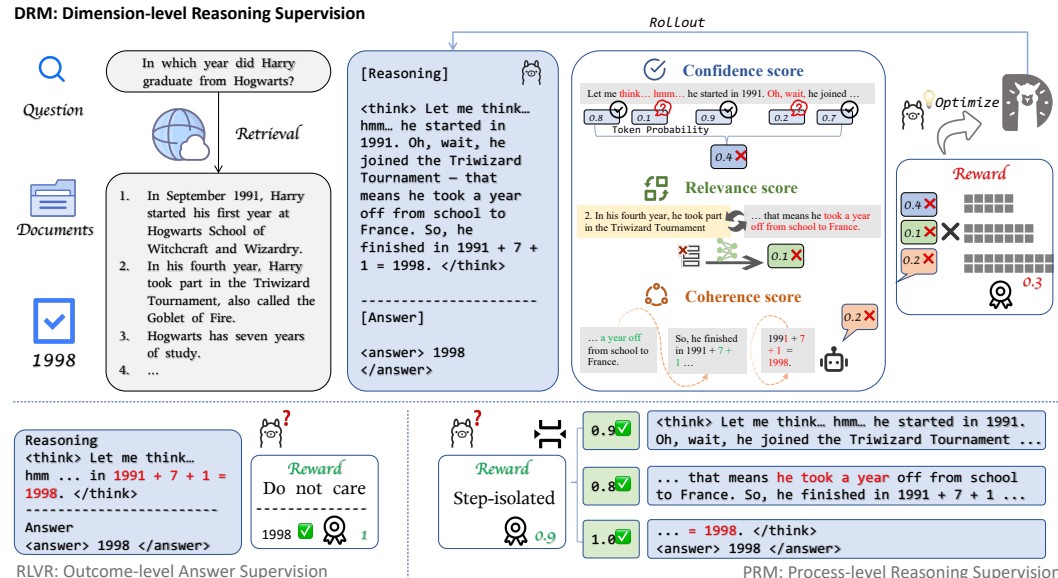

Figure 1: An overview of our multidimensional reasoning supervision framework, illustrated on a RAG task. RLVR regards a *correct answer with flawed reasoning* as a positive sample since it focuses solely on the answer. PRMs also misclassify it because process-level supervision ignores errors across steps when each individual step is correct. **DRM** performs dimension-level supervision, detects reasoning flaws, and assigns a reward that reflects the real quality of reasoning process, facilitating further optimization.

the reasoning remains faithful to the question and supporting context, directly counters the *flawed reasoning* issue where models hallucinate or deviate; (2) **Relevance**, evaluates the semantic relatedness and entailment between the reasoning process and the question, the supporting context and the final answer, enabling the detection of deviations from the given information; and (3) **Coherence**, penalizes self-contradictory statements by the logical consistency of the reasoning process.

Figure 1 illustrates how our framework assesses the quality of the reasoning process as a **D**imension-level **R**eward **M**odel (**DRM**) and addresses the limitations of both RLVR and PRMs. Table 1 summarizes the key properties of the three supervision approaches. By providing a dense, reasoning-aware reward signal without requiring task-specific ground truth answers, DRM overcomes the key limitations of RLVR. Simultaneously, it avoids the task-specific segmentation required by PRMs and offers superior interpretability by scoring reasoning along explicit, diagnosable dimensions.

Table 1: Comparison of supervision approaches.

| Property | RLVR | PRM | **DRM** |
|---|---|---|---|
| Supervision level | Outcome | Process | Dimension |
| Supervision target | Answer | Reasoning | Reasoning |
| Dense signal | ✗ | ✓ | ✓ |
| Generalization | ✓ | ✗ | ✓ |
| Interpretability | ✓ | ✗ | ✓ |
| Ground truth free | ✗ | ✓ | ✓ |

Experimental results on multiple challenging open-domain benchmarks demonstrate the effectiveness of DRM-based supervision in both off-policy selection and on-policy training paradigms. Our results show that DRM-supervised models perform competitively on both in-distribution and out-of-distribution tasks, indicating stronger generalization than answer-supervised counterparts. For LLAMA-3.1-8B-INSTRUCT (Grattafiori et al., 2024), our method achieves performance gains on MATH500 (+8.8, mathematics) (Cobbe et al., 2021a), 2WIKI_RAG (+8.7, multi-hop QA) (Ho et al., 2020) and CRUXEVAL (+7.1, code execution) (Gu et al., 2024). This improvement trend is consistently observed across different models, which unequivocally demonstrates the superiority and generality of DRM supervision. Qualitative analysis and case studies show that DRM mitigates *the correct answer with flawed reasoning* issue common in answer supervision. Our results indicate that multidimensional reasoning supervision enhances the reasoning ability of LLMs and their performance on out-of-distribution tasks.

Table 2: Reasoning assessment dimensions, following the $(Q, D, R, A)$ quadruple format.

| Dimension | Description | Implementation |
|---|---|---|
| **Confidence** $\text{score}^{Conf}$ | Self-assessed certainty of generated $R$ and $A$ from intrinsic signals. | $\text{score}_R^{Conf} = \frac{1}{|R|} \sum \log p$, for all tokens in $R$. $\text{score}_A^{Conf} = \sum \log p$, for all tokens in $A$. $\text{score}^{Conf} = \text{score}_R^{Conf} + \text{score}_A^{Conf}$. |
| **Relevance** $\text{score}^{Rel}$ | Evaluates whether $R$ is contextually appropriate and semantically aligned with $Q$, $D$ and $A$. | $R \leftarrow Q$: Measured by NLI entailment. $R \leftrightarrow D$: Measured by semantic relevance. $R \rightarrow A$: Measured by NLI entailment. |
| **Coherence** $\text{score}^{Coh}$ | Evaluates logical consistency, fluency and overall quality of $R$. | Evaluated by an external ORM. |

## 2 METHODOLOGY: MULTIDIMENSIONAL REASONING SUPERVISION

***Task Definition.*** Formally, let $I$ denote the user input and $O$ the model output. We decompose $O$ into a reasoning process $R$ and an answer $A$. In open-domain scenarios, $I$ often contains more than just the question $Q$. For example, in Retrieval-Augmented Generation (RAG) tasks, $I$ additionally includes retrieved documents, while in preference tasks, $I$ may consist of two candidate responses for the model to compare. Let $D$ denote the additional information accompanying $Q$ and we can decompose $I$ into $Q$ and $D$. Consequently, the input–output structure of the model can be denoted by a quadruple: $(Q, D, R, A)$. In most tasks, the performance of the model is evaluated primarily based on the quality of $A$.

Prior work shows that LLMs sometimes generate unsupported statements during reasoning, which can hinder the production of correct answers (Zhang et al., 2025b; Xu et al., 2025). To address this issue, models are expected to produce faithful reasoning that avoids unsupported claims. In particular, they should produce decisive output, especially for the final answer. Furthermore, the reasoning process should be grounded in the provided input and exhibit internal consistency throughout. These properties support both the production of correct answers and the interpretability of reasoning process. We categorize these properties into three dimensions that a high-quality reasoning process should satisfy: **Confidence**, **Relevance** and **Coherence**. Table 2 summarizes their definitions and implementation and the rationale for each is discussed in the following.

**Confidence.** This dimension evaluates whether the models are certain about their output. Inspired by prior work on self-confidence evaluation in reasoning models, we compute the average log-probability of tokens in $R$ (Leang et al., 2025) to avoid penalizing exploratory reasoning processes. For $A$, we compute the sum of log-probability instead to encourage decisive and confident outputs. The final confidence score is calculated as the sum of these two components.

**Relevance.** This dimension assesses whether $R$ maintains necessary relationships with other components $Q$, $D$ and $A$: (1) $Q \rightarrow R$ should hold via Natural Language Inference (NLI) entailment, ensuring $R$ contributes to answering $Q$; (2) $R \leftrightarrow D$ should exhibit high semantic relevance, ensuring $R$ is grounded in the additional information $D$; and (3) $R \rightarrow A$ should also hold via NLI entailment, ensuring $R$ logically leads to $A$. Specifically, we compute the relevance score by framing it as a ranking task: we rank the reasoning process using three distinct metrics, each corresponding to one of the relationships defined earlier, and then combine these scores to obtain the final score.

**Coherence.** This dimension evaluates the text quality of the reasoning process, with attention to coherence and logical consistency. We treat $R$ as the output of a text generation task with the input of $Q$, $D$. To assess its logical consistency, fluency, and overall textual quality, we use an external Outcome-level Reward Model (ORM) in the text-quality evaluation. This captures another dimension of reasoning quality that is not directly reflected in confidence or relevance.

Overall, by jointly evaluating the reasoning process along **Confidence**, **Relevance** and **Coherence**, our framework explicitly decomposes assessment into complementary dimensions. As illustrated in Figure 1, DRM assesses reasoning quality along three distinct dimensions with each grounded in measurable scores. We compute the DRM reward by a weighted sum of the dimensional scores:

$$R_i^{DRM} = \text{score}_i = \sum_D w^D \widetilde{\text{score}}_i^D, \quad D \in \{\text{Conf}, \text{Rel}, \text{Coh}\},$$

where $\widetilde{\text{score}}_i^D$ is the component $\text{score}_i^D$ after being individually normalized within its group to mitigate scale differences. This produces a dense reward that serves as a direct supervision signal. The weights are determined via a grid search on the validation set. This design inherently avoids the binary sparse reward issue of RLVR and reflects the quality of the reasoning process. DRM replaces stepwise scoring with dimension-wise assessment and eliminates the need for task-specific step segmentation in PRMs. Owing to its dimensional nature, DRM inherently provides more interpretable feedback. Moreover, it can distinguish among multiple reasoning processes by their quality, regardless of answer correctness. As DRM addresses the evaluation limitations of RLVR and PRM, we investigate whether its reward can serve as an effective supervision signal for LLM optimization. In off-policy optimization, training sets are constructed under the guidance of a supervision signal. $R_i^{DRM}$ can serve this role by capturing the reasoning quality of each sample, thereby facilitating training set construction. We adopt DPO, and its optimization objective is formulated as follows:

$$\mathcal{L}_{\text{DPO}}(\theta) = -\mathbb{E}_{(I,O^+,O^-)}\left[\log \sigma\left(\beta \log \frac{\pi_\theta(O^+ \mid I)}{\pi_{\text{ref}}(O^+ \mid I)} - \beta \log \frac{\pi_\theta(O^- \mid I)}{\pi_{\text{ref}}(O^- \mid I)}\right)\right],$$
$$O^+ = \arg\max_{o \in O} R_o^{DRM}, \; O^- = \arg\min_{o \in O} R_o^{DRM},$$

where $\sigma(\cdot)$ is the sigmoid function and $\beta > 0$ controls the sharpness of preference. In on-policy optimization, DRM can serve as a standalone supervision reward signal, or be integrated with other supervision signals. Specifically, we compute an additional DRM advantage $\hat{A}_{i,t}^{DRM}$ from $R_i^{DRM}$, which denotes the DRM reward for sample $i$. We then add this DRM advantage to the native GRPO advantage $\hat{A}_{i,t}$ obtained from RLVR rewards, yielding our optimization objective (for mathematical details, please refer to Appendix B.2):

$$\mathcal{J}_{\text{GRPO}}(\theta) = \mathbb{E}_{q,\{o_i\}} \frac{1}{G} \sum_{i=1}^{G} \frac{1}{|o_i|} \sum_{t=1}^{|o_i|} \left\{ \min\left[r_{i,t}(\theta)A_{i,t}, \text{clip}(r_{i,t}(\theta), 1-\varepsilon, 1+\varepsilon)A_{i,t}\right] \right.$$
$$\left. - \beta \, \mathbb{D}_{\text{KL}}\left[\pi_\theta \| \pi_{\text{ref}}\right] \right\}, \quad A_{i,t} = \begin{cases} \hat{A}_{i,t}, & \text{RLVR}, \\ \hat{A}_{i,t}^{DRM}, & \text{DRM}, \\ \hat{A}_{i,t} + \hat{A}_{i,t}^{DRM}, & \text{Combination of RLVR and DRM}, \end{cases}$$

where $r_{i,t}(\theta) = \frac{\pi_\theta(o_{i,t}|q,o_{i,<t})}{\pi_{\theta_{\text{old}}}(o_{i,t}|q,o_{i,<t})}$ is the token-level probability ratio and $\beta$ controls the KL penalty strength with respect to a reference policy $\pi_{\text{ref}}$. DRM can be employed either as a standalone signal or integrated with the RLVR supervision signal.

## 3 Experiments

Following a rigorous experimental paradigm, we formulate a set of research questions (RQs) to evaluate whether DRM supervision can improve the model's reasoning ability. The empirical results presented in this section affirmatively answer all of the following research questions.

**RQ1:** *Can assessment on reasoning process reliably determine the final answer correctness?*
**RQ2:** *Can the DRM reward signal be learned and used by models to improve reasoning ability?*
**RQ3:** *Can DRM supervision better guide training and outperform RLVR?*
**RQ4:** *Can combining RLVR supervision with DRM supervision lead to further improvements?*

### 3.1 Experimental Setup

**Models.** We evaluate our method on three representative models: a model lacking inherent reasoning ability **Llama-3.1-8B-Instruct** (Grattafiori et al., 2024), a reasoning model **R1-distil-Llama8B** (DeepSeek-AI et al., 2025), and a hybrid reasoning model **Qwen3-8B** (Yang et al., 2025). We employ Qwen3-8B-reranker (Zhang et al.) as the relevance judge and Llama-3.3-Nemotron-70B-Reward-Multilingual (Wang et al.) as the coherence judge.

Table 3: Answer correctness (%) of DRM construction approaches on RewardBench2. Native means the performance of the backbone models. (0.1,0.2,0.7) means weights for Confidence, Relevance and Coherence are 0.1, 0.2, 0.7, respectively. LTR denotes the use of a Learning-to-Rank model with learnable weights for integration. The highest result in each row is in **bold**.

| Model | Native | Confidence | Relevance | Coherence | Weighted Equally | Weighted (0.1,0.2,0.7) | LTR |
|---|---|---|---|---|---|---|---|
| LLaMA3.1-8B-Instruct | 67.17 | 65.44 | 72.32 | 78.55 | 77.45 | 78.57 | **79.13** |
| R1-Distil-Llama8B | 63.46 | 63.10 | 66.76 | **76.35** | 75.11 | 76.16 | 75.18 |
| Qwen3-8B | 84.87 | 83.20 | 85.10 | 85.54 | 85.01 | 85.65 | **85.88** |

**Datasets.** We evaluate our method on a diverse set of open-domain tasks, including four **Code** benchmarks, two **Preference** benchmarks, four **Math** benchmarks, two **Scientific QA** benchmarks, three **Logical Reasoning** benchmarks and two **Question Answering** benchmarks along with their RAG variants provided by FlashRAG (Jin et al., 2024). For math tasks, we use **MATH-VERIFY** (Kydlíček, 2024) for automatic solution verification and **exact match** for all other tasks.[1]

## 3.2 EVALUATING WHETHER DRM GUIDES CORRECT ANSWERS

To address **RQ1**, we validate the effectiveness of DRM using a Best-of-N (BoN) selection setup. The underlying hypothesis is that a high-quality reasoning process assessed by our multi-dimensional reward serves as a reliable proxy for answer correctness. Specifically, for each test instance, we sample multiple candidate reasoning paths from the model and select the one with the highest DRM reward. We then evaluate whether this selection mechanism yields higher answer accuracy compared to three types of baselines: a baseline obtained via uniform sampling of reasoning processes, which reflects the model's native performance in the absence of explicit supervision signals; baselines using each individual DRM dimension (**Confidence**, **Relevance**, or **Coherence**) in isolation, which allows us to assess the contribution of each signal separately; and a baseline where these three dimensions are integrated with equal weights. Furthermore, we also compare fixed weighting schemes against learnable weights. We employ a Learning-to-Rank (LTR) approach based on LambdaRank (Burges et al., 2007; Burges, 2010), training the model to optimize the combination of dimensional scores to maximize the probability of correctness.

As shown in Table 3, DRM consistently achieves higher accuracy than the backbone models. While using the **Confidence** score alone slightly reduces accuracy, combining it with **Relevance** and **Coherence** improves performance, indicating that these dimensions capture complementary aspects of reasoning quality. Regarding integration mechanisms, the combined approach consistently outperforms both individual metrics and native backbone performance, regardless of whether the integration employs equal weighting, grid-search fixed weights or a learnable mechanism. This stability is observed across diverse backbones and is further validated on a distinct data distribution (HotpotQA with RAG) in Table 11. Given that the performance gap between fixed weights and the more complex LTR approach is marginal, we determine the combination weights via grid search on the validation set and fix them for all subsequent experiments. This choice prioritizes simplicity and robustness, eliminating the need for additional training to learn parameters. Overall, the results of our extensive experiments demonstrate that DRM maintains robustness across different backbone models, integration methods and training data distributions.

## 3.3 ASSESSING THE EFFECTIVENESS OF DRM SUPERVISION

This section focuses on **RQ2** and **RQ3**. We conduct off-policy reinforcement learning using DPO with Supervised Fine-Tuning (SFT) loss (for mathematical details, please refer to Appendix B.1). We construct separate training sets based on different supervision signals. Specifically, DRM rewards serve as reasoning supervision signals, guiding the selection of samples with higher reasoning quality, while RLVR rewards serve as answer supervision signals, selecting samples based on answer correctness. For each instance in RewardBench2, we prompt the model to generate 20 sam-

---

[1]The main paper only reports results on RewardBench2; results for HotpotQA with RAG are provided in Appendix G.

ples containing step-by-step reasoning and final answers. These samples are scored and selected according to the respective supervision signal to form preference pairs, as described below.

---

**Training Set Construction.**
Let $x$ denote a sample from set $X$, where all samples in $X$ are generated from the same instance. Each sample is associated with a correctness label $\text{answer}_x \in \{\text{True}, \text{False}\}$ and a reasoning quality $\text{score}_x$. The positive set $X^+$ and negative set $X^-$ are defined according to a **SUBSET** rule and preference pairs are selected according to a **SUPERVISION** method. Once these two components are specified, the resulting training set is uniquely determined.
**SUBSET**:
    **ANY**: $X^+ = X^- = X$.
    **T+T**: $X^+ = X^- = \{x \mid \text{answer}_x = \text{True}, x \in X\}$.
    **T+F**: $X^+ = \{x \mid \text{answer}_x = \text{True}, x \in X\}$,    $X^- = \{x \mid \text{answer}_x = \text{False}, x \in X\}$.
    **F+F**: $X^+ = X^- = \{x \mid \text{answer}_x = \text{False}, x \in X\}$.
**SUPERVISION**
    **DRM**: $\{(x^+, x^-) \mid x^+ = \arg\max_{x \in X} \text{score}_x, \ x^- = \arg\min_{x \in X} \text{score}_x\}$
    **RLVR**: $\{(x^+, x^-) \mid x^+ = \text{random}(X^+), \ x^- = \text{random}(X^-)\}$
Let **SUPERVISION@SUBSET** denote a training set construction method. For example, **DRM@T+F** indicates that we select a sample with the highest DRM reward and correct answer and pair it with a sample with the lowest DRM reward and wrong answer. It is clear that **DRM@ANY** refers to the training set constructed with DRM supervision. In contrast, **RLVR@T+F** refers to the training set constructed with answer supervision, under the RLVR assumption that samples with the same answer are considered equivalent.

---

We construct separate training sets and train models on each set independently. The full training details are provided in Appendix F.3. As shown in Table 4, DRM-supervised training consistently outperforms RLVR-supervised training, providing evidence in support of both research questions.

**RQ2.** To assess whether DRM reward signals can be effectively learned and used to improve reasoning ability, we compare **NATIVE** and **DRM@ANY**. Additionally, we include **RLVR@ANY** as a control group, in which the training set was constructed randomly. In the **DRM@ANY** setting, the training set is constructed entirely based on DRM reward signals, without incorporating any information about answer correctness. Table 4 shows that **DRM@ANY** achieves higher scores than all other settings, with substantial improvements across all evaluated datasets. The strong performance on out-of-distribution tasks suggests that the model generalizes well beyond the training distribution. The results indicate that the proposed DRM supervision can be effectively learned even without answer supervision, i.e., without access to the ground truth answers.

**RQ3.** We compare DRM and RLVR across two key aspects to assess their relative effectiveness:
*Performance gain:* To evaluate the effectiveness of DRM, we compare **RLVR@T+F** with **DRM@ANY** (see Table 4). This comparison examines whether explicit supervision of reasoning achieves better performance than supervising only the answer. In this setting, **DRM@ANY** consistently achieves higher performance than **RLVR@T+F**, indicating that training with DRM supervision consistently outperforms RLVR supervision.
*Overcoming limitations:* We compare **RLVR@T+T** with **DRM@T+T** and **RLVR@F+F** with **DRM@F+F** to test whether DRM can still provide supervision when all answers have identical correctness labels, where RLVR cannot produce a preference signal. Results show that DRM can distinguish reasoning quality in such case, demonstrating its ability to generate informative supervision and to enhance the model's ability to handle a broader range of scenarios.

Furthermore, we conduct off-policy training and compare it against the baselines as shown in Table 5. We evaluate our model against three strong baselines: (1) a model trained on the **ANY** subset with reasoning supervision signals from SKYWORK-REWARD-V2-LLAMA-3.1-8B, a powerful ORM, (2) RLPR (Yu et al., 2025b) and (3) KLEAR (Su et al., 2025). Both RLPR and KLEAR are reasoning-enhanced models trained using the same backbone architecture as their counterparts in our experiments. This setup allows us to examine whether our DRM provides more effective and generalizable supervision than existing reasoning-supervision approaches. We also examines whether DRM-supervised models can outperform models optimized with other methods. Across most downstream open-domain tasks, DRM outperforms all three baselines. In particular, it surpasses RLPR and KLEAR under the same backbone, demonstrating its effectiveness. It also exceeds the perfor-

Table 4: Results of controlled comparisons for RQ2 and RQ3. We use LLAMA3.1-8B-INSTRUCT as the base model. Results for other models, which exhibit the same trend, are provided in Appendix G.2. As described in Section 3.1, we use MATH-VERIFY as the evaluation metric for math tasks and EM for all other tasks, respectively. All models are trained for the same number of steps to ensure a fair comparison. For each row within a comparison, the highest score is in **bold**.

| Task Domain | Dataset | | For RQ2, RQ3.1 | | | For RQ3.2 | | | |
|---|---|---|---|---|---|---|---|---|---|
| | | Native | RLVR @ANY | RLVR @T+F | DRM @ANY | RLVR @T+T | DRM @T+T | RLVR @F+F | DRM @F+F |
| Code | CodeMMLU | 58.8 | 58.8 | 59.5 | **59.9** | 58.9 | **59.6** | 59.6 | **61.3** |
| | CodeScope | 34.8 | 35.4 | 37.4 | **41.1** | 36.2 | **41.0** | 36.6 | **40.0** |
| | Cruxeval | 50.4 | 53.5 | 52.6 | **57.5** | 53.6 | **56.6** | 53.9 | **55.9** |
| | Execution-v2 | 38.2 | 40.9 | 43.2 | **45.3** | 39.2 | **45.5** | 40.3 | **46.8** |
| Preference | RM-Bench | 56.4 | 59.3 | 59.2 | **61.0** | 60.0 | **60.3** | 59.7 | **61.9** |
| | UltraFeedback | 66.6 | 65.6 | 65.4 | **69.9** | 66.4 | **67.7** | 64.5 | **68.8** |
| Math | AIME24 | 4.7 | 4.7 | 4.0 | **6.0** | 4.7 | **7.3** | **4.7** | 4.0 |
| | AMC23 | 22.5 | 23.5 | 23.5 | **29.5** | 23.0 | **25.5** | 22.0 | **26.5** |
| | GSM8K | 88.8 | 89.0 | 89.5 | **91.8** | 90.2 | **91.7** | 88.7 | **91.7** |
| | Math500 | 39.6 | 41.4 | 43.4 | **48.4** | 42.0 | **46.6** | 40.0 | **48.4** |
| Scientific QA | MMLU-Pro | 41.9 | 45.3 | 46.4 | **48.7** | 45.7 | **48.4** | 46.6 | **49.0** |
| | GPQA | 31.3 | 28.8 | 32.8 | **35.9** | 29.8 | **30.3** | 29.8 | **35.4** |
| Reasoning | MuSR | 48.3 | 49.5 | 49.7 | **51.7** | 48.3 | **53.3** | 49.7 | **51.6** |
| | DROP | 56.9 | 61.0 | 62.9 | **63.6** | 60.0 | **64.4** | 58.5 | **65.1** |
| | QASC | 84.4 | 84.0 | 84.2 | **87.2** | 83.8 | **87.8** | 83.4 | **86.2** |
| QA | 2wiki | 33.8 | 33.2 | 34.6 | **35.6** | 32.3 | **32.7** | 30.7 | **33.4** |
| | HotpotQA | 29.3 | 29.9 | 30.1 | **31.8** | 29.3 | **30.1** | 29.1 | **29.7** |
| QA-RAG | 2wiki_RAG | 31.2 | 32.1 | 35.8 | **39.9** | 36.6 | **41.4** | 32.1 | **43.3** |
| | HotpotQA_RAG | 28.3 | 28.3 | 32.3 | **34.5** | 29.3 | **32.3** | 28.5 | **33.8** |

mance of the model trained with SKYWORK supervision, indicating that DRM consistently achieves stronger and more generalizable reasoning ability. The improvements are consistent across various architectures and tasks, suggesting that DRM is an architecture-agnostic approach that generalizes well. Notably, our training relies solely on preference data from RewardBench2, the same type of data used for training reward models (Zhang et al., 2025a; Zhong et al., 2025), without access to ground truth answers or task-specific finetuning. This highlights the data efficiency of our approach as a single source of preference data leads to broad improvements across open-domain tasks.

## 3.4 ENHANCING RLVR WITH DRM

This section addresses **RQ4**. We conduct on-policy GRPO training on three advantage configurations: answer supervision only, reasoning supervision only and their combination. This setup directly tests whether DRM supervision and integrating DRM rewards into RLVR achieve further gains. The comparison between RLVR and DRM also examines whether the trend observed in off-policy training remains consistent in on-policy stages. GRPO training details are provided in Appendix F.4.

Across most model backbones and representative benchmarks on open-domain tasks, the combined approach performs as well as or better than the best single supervision approach, as shown in Table 6. This trend is also consistently observed in the off-policy setting. The combination also outperforms RLVR, indicat-

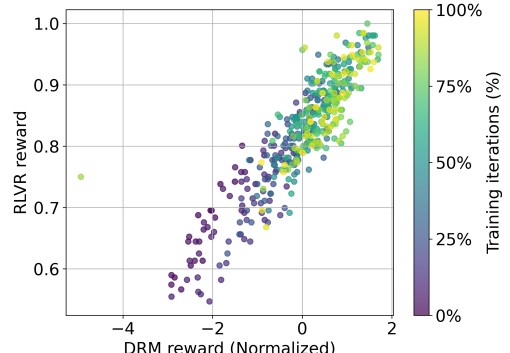

Figure 2: The relationship between RLVR rewards and DRM rewards in R1-Distil-Llama8B Combination training. Each data point represents a single training batch. Note that DRM rewards are Z-score normalized for better visualization.

Table 5: Results of off-policy DPO with SFT loss training. **RLPR** and **KLEAR** are baseline models that share the same backbone architectures as their respective counterparts. **SKYWORK** indicates that the model's training set is constructed using SKYWORK reward model. **DRM** represents **DRM@ANY**. For each row within a model group, the highest score is in **bold**.

| Task Domain | Dataset | LLaMA3.1-8B-Instruct | | | | R1-Distil-Llama8B | | | Qwen3-8B | | | |
| | | Native | RLPR | SKYWORK | DRM | Native | SKYWORK | DRM | Native | KLEAR | SKYWORK | DRM |
|---|---|---|---|---|---|---|---|---|---|---|---|---|
| Code | CodeMMLU | 58.8 | 58.0 | 57.6 | **59.9** | 59.7 | 62.9 | **66.3** | 77.9 | 77.4 | 79.3 | **80.3** |
| | CodeScope | 34.8 | 38.7 | 39.3 | **41.1** | 67.4 | 68.2 | **70.2** | 86.5 | **88.1** | 86.2 | 87.4 |
| | Cruxeval | 50.4 | 53.6 | 53.6 | **57.5** | 71.9 | 77.0 | **77.2** | 91.6 | 87.2 | 91.9 | **93.0** |
| | Execution-v2 | 38.2 | 44.7 | 42.8 | **45.3** | 80.8 | 82.0 | **86.0** | 98.5 | 95.2 | 97.9 | **99.0** |
| Preference | RM-Bench | 56.4 | 60.2 | 59.8 | **61.0** | 71.9 | 73.4 | **74.6** | 85.4 | 83.7 | 85.1 | **85.6** |
| | UltraFeedback | 66.6 | 68.5 | 67.0 | **69.9** | 65.2 | 66.5 | **66.8** | 71.3 | 68.1 | 72.2 | **73.2** |
| Math | AIME24 | 4.7 | **6.0** | 4.0 | **6.0** | 28.7 | 26.7 | **33.3** | 38.0 | 40.0 | 38.7 | **44.7** |
| | AMC23 | 22.5 | 26.0 | 25.5 | **29.5** | 70.5 | 74.5 | **75.5** | 72.0 | 75.0 | 76.0 | **79.0** |
| | GSM8K | 88.8 | 90.0 | 89.8 | **91.8** | 66.7 | **73.7** | 69.2 | 95.6 | 93.8 | 95.8 | **96.1** |
| | Math500 | 39.6 | 47.2 | 42.6 | **48.4** | 62.6 | **65.6** | 63.2 | 73.2 | 68.2 | 72.6 | **75.6** |
| Scientific QA | MMLU-Pro | 41.9 | 36.3 | 46.7 | **48.7** | 51.5 | 52.8 | **54.7** | 65.3 | 67.1 | 70.0 | **71.4** |
| | GPQA | 31.3 | 30.8 | 33.3 | **35.9** | 39.9 | 37.4 | **44.9** | 48.0 | 55.6 | 52.5 | **58.1** |
| Reasoning | MuSR | 48.3 | 48.7 | 49.7 | **51.7** | 52.6 | 52.8 | **54.1** | 63.5 | 50.8 | 63.5 | **65.5** |
| | DROP | 56.9 | 45.4 | 63.0 | **63.6** | 50.8 | **54.5** | 50.2 | 74.7 | 68.8 | 74.2 | **74.9** |
| | QASC | 84.4 | 87.0 | 87.1 | **87.2** | 82.1 | 82.5 | **84.1** | 94.1 | 93.3 | 93.7 | **94.2** |
| QA | 2wiki | 33.8 | 32.1 | 32.4 | **35.6** | 26.2 | 29.3 | **31.6** | 39.8 | 35.9 | 40.0 | **42.2** |
| | HotpotQA | 29.3 | 29.9 | 30.4 | **31.8** | 18.1 | 19.3 | **19.7** | 29.2 | 19.6 | 29.1 | **29.4** |
| QA-RAG | 2wiki_RAG | 31.2 | 38.7 | 34.8 | **39.9** | 36.7 | **39.2** | 37.9 | 55.7 | 52.2 | 55.8 | **56.1** |
| | HotpotQA_RAG | 28.3 | 32.8 | 33.2 | **34.5** | 27.1 | 26.5 | **27.3** | 40.5 | 34.3 | 40.3 | **40.7** |

ing that incorporating reasoning supervision alongside answer supervision consistently improves performance by guiding intermediate reasoning steps during policy optimization. When compared to DRM, the combination yields gains, but shows slight drops in certain reasoning-focused or knowledge-intensive datasets, such as MuSR and GPQA, suggesting that in these cases direct RLVR may interfere with the optimization due to overlooking the reasoning process. We provide empirical evidence for this interference in Figure 2, illustrating the correlation between DRM and RLVR rewards throughout the Combination method training iterations. While there is a positive global trend, the outliers indicate that the two reward signals are not always synchronized. These outliers represent conflicting supervision signals, which can cause the combination method to underperform compared to the pure process-level supervision provided by DRM. Overall, these findings indicate that integrating answer and reasoning supervision provides stable improvements across diverse open-domain tasks, supporting an affirmative answer to **RQ4**.

## 4 ANALYSIS

### 4.1 CAN DRM LEAD TO HIGH-QUALITY REASONING PROCESS?

As introduced in Section 2, most tasks are evaluated solely based on answer correctness, regardless of the quality of the reasoning process that produced the answer. However, a clear and coherent reasoning process helps users assess and trust the output in interactions with LLMs. This section examines whether DRM can identify truly high-quality reasoning process. We prompt GPT-4o to determine whether a reasoning process and its corresponding answer constitute a *correct answer with flawed reasoning* in off-policy training sets constructed with two different supervision approaches. In these settings, RLVR denotes answer supervision while DRM denotes reasoning supervision. As shown in Figure 3a, the number of *correct answer with flawed reasoning* instances decreases substantially across all models when using DRM.

Furthermore, we investigate whether DRM supervision leads to more structured reasoning patterns. As shown in Figure 3b, our analysis reveals that models trained with DRM exhibit improved structural coherence, producing solutions that are not only logically sound but also more organized and systematic compared to backbone models.

Table 6: Results of on-policy GRPO training. **RLVR** denotes training with answer supervision only. **DRM** denotes training with reasoning supervision only. **Combination** denotes training with their combination. Only representative benchmarks are reported here for brevity, with complete results in Appendix G.3. For each row within a model group, the highest score is in **bold**.

| Task Domain | Dataset | LLaMA3.1-8B-Instruct | | | R1-Distil-Llama8B | | | Qwen3-8B | | |
|---|---|---|---|---|---|---|---|---|---|---|
| | | RLVR | DRM | Combination | RLVR | DRM | Combination | RLVR | DRM | Combination |
| Code | CodeScope | 37.2 | 39.4 | **40.5** | 69.2 | 68.2 | **70.8** | 87.3 | **87.7** | 87.5 |
| | Execution-v2 | 44.7 | 42.4 | **46.4** | 82.3 | 83.5 | **85.6** | 98.5 | 99.0 | **99.2** |
| Math | AIME24 | **4.7** | **4.7** | **4.7** | 29.3 | **34.7** | 33.3 | 38.0 | **46.7** | 45.3 |
| | AMC23 | 20.5 | 23.0 | **24.5** | 70.5 | 77.5 | **80.5** | 75.0 | **81.5** | 79.5 |
| | Math500 | 40.8 | 38.0 | **45.4** | 62.8 | 67.0 | **67.2** | 73.8 | **75.8** | **75.8** |
| Scientific QA | MMLU-Pro | 42.3 | 43.2 | **47.8** | 53.6 | 53.4 | **54.1** | 63.7 | 68.7 | **69.1** |
| | GPQA | 30.8 | 28.8 | **32.3** | 39.4 | **43.9** | 42.4 | 43.9 | **57.6** | 56.6 |
| Reasoning | MuSR | 47.6 | **52.9** | 52.1 | **53.0** | **53.0** | 52.9 | 63.0 | 63.2 | **64.3** |

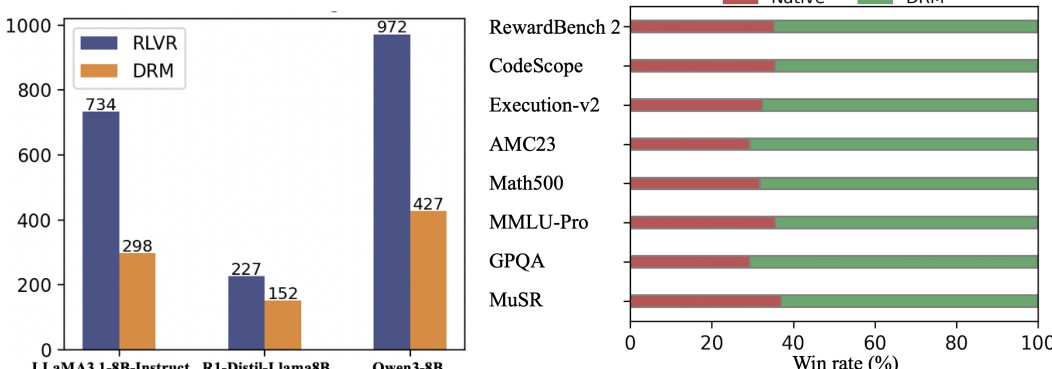

(a) Count of *correct answers with flawed reasoning* as evaluated by GPT-4o. Each training set contains approximately 6,000 samples.

(b) Comparison of reasoning process structure between R1-Distil-Llama8B and its DRM-supervised variants as evaluated by GPT-4o. Note that ties are excluded from the plot.

Figure 3: Analysis of DRM supervision effectiveness: (a) reduction in flawed reasoning cases; (b) lead to more structured reasoning process.

These results indicate that DRM prioritizes instances with higher reasoning quality compared to RLVR, confirming that reasoning supervision successfully identifies real high-quality reasoning process associated with completely correct answers. Together with the experiments addressing **RQ1** in Section 3.2, we demonstrate that our multidimensional reasoning supervision not only produces more correct answers but also improves reasoning quality by reducing *correct answer with flawed reasoning* and enhancing structural organization.

## 4.2 ABLATION STUDY OF INDIVIDUAL SUPERVISION DIMENSIONS

We conduct an ablation study to examine the effect of each reasoning supervision dimension in isolation. Starting from the native model, we adopt the same off-policy training setting and apply supervision to only one dimension at a time: **Confidence**, **Relevance**, or **Coherence**, while keeping all other training settings fixed. As shown in Figure 4, supervision of a single dimension yields improvements on some specific tasks but can also lead to performance drops on others. This pattern suggests that each dimension captures a distinct aspect of the model's reasoning ability and tends to excel at different types of tasks. No single dimension is sufficient on its own for robust improvements across diverse tasks. In contrast, combining multiple complementary dimensions (DRM) produces cooperative effects that leverage the strengths of each dimension and enhance the model's generalization ability. This combination achieves broader and more consistent gains, which cannot be attributed to any single dominant dimension.

## 5 RELATED WORK

### 5.1 REINFORCEMENT LEARNING WITH VERIFIABLE REWARDS

RLVR effectively improves LLM reasoning ability (DeepSeek-AI et al., 2025; Team et al., 2025; Yang et al., 2025) by using automatically verifiable correctness signals as rewards, guiding models to explore reasoning trajectories that produce correct solutions (Lambert et al., 2025; Zhang et al., 2025b; OpenAI et al., 2024). Shao et al. (2024) introduce GRPO as an optimization method for RLVR. GRPO is a variant of Proximal Policy Optimization (PPO) (Schulman et al., 2017) that replaces the separate value function with a group-based relative advantage estimation, removing the need for an additional critic model and enabling large-scale training (Shao et al., 2024).

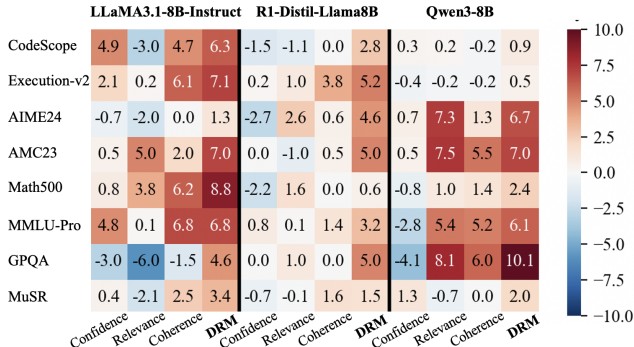

Figure 4: Ablation results of single dimension supervised training. The values in this heatmap indicate the absolute difference relative to the native model. Training pairs are selected from **ANY** subset. **DRM** means training with DRM supervision.

### 5.2 REWARD MODELS

**Outcome-level Reward Models** Given a user input, ORMs assess the corresponding model response and assign a score reflecting its outcome-level quality (Zhang et al., 2025b; Zhong et al., 2025). They are typically trained on preference datasets and have been applied to a range of open-domain tasks (Liu et al., 2025; Zhong et al., 2025; Liu et al., 2025; Wang et al.). Since ORMs evaluate the overall response, they may assign high scores to answers that are correct but obtained through flawed reasoning, as they do not explicitly assess the reasoning process (Lightman et al., 2024; Cheng et al., 2025; Wang et al., 2025).

**Process-level Reward Models** PRMs are designed to evaluate the reasoning process rather than only the final answer. OpenORM (Zhang et al., 2025a) extends an LLM into a PRM for pairwise open-domain evaluation, which can limit efficiency when used as a training reward (Zhong et al., 2025). Pointwise PRMs, such as ReasonFlux-PRM (Zou et al., 2025), assign scores to individual intermediate steps in a reasoning trace, often relying on learned task-specific segmentation patterns. ROSCOE (Golovneva et al., 2023) and ReCEval (Prasad et al., 2023) investigate methods for evaluating the quality of chain-of-thoughts. These approaches focus on scoring the reasoning process but lack empirical validation of whether such signals can be effectively learned by models.

## 6 CONCLUSION

In this paper, we present a multidimensional reasoning-level supervision framework. It can automatically assess the reasoning quality of LLMs without ground truth answers, aggregating **Confidence**, **Relevance** and **Coherence** into a dense and interpretable score. Our framework serves as a dimension-level reward model that directly reflects the quality of reasoning process. DRM provides dense and reasoning-aware supervision signals without requiring step segmentation, thereby addressing key limitations of both RLVR and PRMs. We show that **DRM** rewards can be applied in both off-policy preference optimization and on-policy reinforcement learning and can be combined with verifiable answer rewards to jointly improve reasoning quality and answer correctness. Experiments on diverse open-domain tasks demonstrate consistent improvements in in-distribution and out-of-distribution settings, highlighting the effectiveness and generality of our supervision approach. Notably, these improvements are achieved without task-specific data or training, highlighting the data efficiency of our framework. We anticipate that the insights gained from our study of multidimensional reasoning supervision will lay a solid foundation for future research aimed at enhancing both the interpretability and generalization of LLM reasoning ability.

ETHICS STATEMENT

This study is based on publicly available datasets and does not involve any personally identifiable or sensitive information.

REPRODUCIBILITY STATEMENT

Codes and scripts are provided in the supplementary materials to reproduce the empirical results. All models and datasets used in our experiments are obtained from the Hugging Face Hub[2].

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

# A  THE USE OF LARGE LANGUAGE MODELS

Large language models were used to help refine the writing of this manuscript. The authors reviewed and verified all content.

# B  MATHEMATICAL DETAILS OF USED METHODS

This section follows the quadruple notation of $(Q, D, R, A)$ defined in Section 2.

## B.1  DPO WITH SFT LOSS

Rafailov et al. (2023) proposes Direct Preference Optimization, a direct approach to align LLMs with human preferences using paired comparison data, without requiring an explicit reward model. Building on prior work (Rafailov et al., 2023; von Werra et al., 2020; Zhao et al., 2025), we additionally incorporate a Supervised Fine-Tuning (SFT) loss to stabilize training. The complete mathematical formulation is presented below.

Given a user input $I$ and two candidate outputs $(O^+, O^-)$, where $O^+$ is preferred over $O^-$, the standard DPO objective optimizes the model parameters $\theta$ by maximizing the log-likelihood ratio between the preferred and dispreferred outputs under the current policy $\pi_\theta$ and a reference policy $\pi_{\text{ref}}$:

$$\mathcal{L}_{\text{DPO}}(\theta) = -\mathbb{E}_{(I, O^+, O^-)} \left[ \log \sigma \left( \beta \log \frac{\pi_\theta(O^+ \mid I)}{\pi_{\text{ref}}(O^+ \mid I)} - \beta \log \frac{\pi_\theta(O^- \mid I)}{\pi_{\text{ref}}(O^- \mid I)} \right) \right], \tag{1}$$

where $\sigma(\cdot)$ is the sigmoid function and $\beta > 0$ controls the sharpness of preference.

Given a set of preferred responses from the DPO training pairs $\mathcal{D}_{\text{SFT}} = \{(I, O^+)\}$, we define:

$$\mathcal{L}_{\text{SFT}}(\theta) = -\mathbb{E}_{(I, O^+) \sim \mathcal{D}_{\text{SFT}}} \left[ \log \pi_\theta(O^+ \mid I) \right] \tag{2}$$

Combining these two losses, we have:

$$\mathcal{L}_{\text{DPO-SFT}}(\theta) = \mathcal{L}_{\text{DPO}}(\theta) + \lambda_{\text{SFT}} \, \mathcal{L}_{\text{SFT}}(\theta), \tag{3}$$

where $\lambda_{\text{SFT}} \geq 0$ is the relative weight of the SFT loss.

## B.2  GRPO

As discussed in Section 5, GRPO replaces the separate value function with a group-based relative advantage estimation. For each question $q$, the policy $\pi_{\theta_{\text{old}}}$ generates $G$ candidate outputs $\{o_i\}_{i=1}^G$. The advantage for each token $o_{i,t}$ is computed as

$$\hat{A}_{i,t} = \frac{R_i - \text{mean}(\{R_j\}_{j=1}^G)}{\text{std}(\{R_j\}_{j=1}^G)}, \tag{4}$$

where $R_i$ denotes the scalar reward assigned to output $o_i$. This formulation normalizes rewards within the group. In the native GRPO implementation, the reward is binary and determined by an automatic rule-based verifier:

$$R_i = \begin{cases} 1, & \text{if the verifier returns } \texttt{true} \text{ for output } o_i, \\ 0, & \text{otherwise.} \end{cases} \tag{5}$$

The GRPO objective is defined as

$$\mathcal{J}_{\text{GRPO}}(\theta) = \mathbb{E}_{q, \{o_i\}} \frac{1}{G} \sum_{i=1}^G \frac{1}{|o_i|} \sum_{t=1}^{|o_i|} \left\{ \min \left[ r_{i,t}(\theta) \hat{A}_{i,t}, \ \text{clip}(r_{i,t}(\theta), 1 - \varepsilon, 1 + \varepsilon) \hat{A}_{i,t} \right] \right.$$
$$\left. - \beta \, \mathbb{D}_{\text{KL}} \left[ \pi_\theta \, \| \, \pi_{\text{ref}} \right] \right\}, \tag{6}$$

where $r_{i,t}(\theta) = \frac{\pi_\theta(o_{i,t} \mid q, o_{i,<t})}{\pi_{\theta_{\text{old}}}(o_{i,t} \mid q, o_{i,<t})}$ is the token-level probability ratio and $\beta$ controls the KL penalty strength with respect to a reference policy $\pi_{\text{ref}}$.

### B.3 GRPO WITH DRM SUPERVISION SIGNALS

In Section 3.4, we assign an additional advantage using DRM supervision signals to GRPO native advantage. Formally, that is:

$$A_{i,t} = \hat{A}_{i,t} + \hat{A}_{i,t}^{DRM}, \tag{7}$$

where $\hat{A}_{i,t}$ is native GRPO loss computed by answer-level verified rewards in Equation 4 and Equation 5. $\hat{A}_{i,t}^{DRM}$ denotes the DRM supervision advantages, computed as:

$$\hat{A}_{i,t}^{DRM} = \sum_D w^D \hat{A}_{i,t}^D \tag{8}$$

$$\hat{A}_{i,t}^D = \frac{R_i^D - \text{mean}(\{R_j^D\}_{j=1}^G)}{\text{std}(\{R_j^D\}_{j=1}^G)} \tag{9}$$

$$R_j^D = \text{score}_i^D, \tag{10}$$

where $\text{score}_i^D$ is the reasoning score of sample $i$.

## C  CASE STUDY

**Analysis.**   The case below is generated by LLAMA3.1-8B-INSTRUCT and illustrates the *correct answer with flawed reasoning problem* in answer supervision. The first assistant response is randomly sampled from the set of correct answers. The second is selected by DRM. Both samples correctly choose Response 1 as the better one. In this case, the sampled response chooses Response 1, which follows the privacy policy. However, its reasoning overlooks the privacy violation in Response 2 and treats "providing some information" as a positive factor. This results in an inconsistent explanation for the choice and reflects the faults in its reasoning process. The DRM selected response, in contrast, identifies the privacy risk and the possible inaccuracy in Response 2. It also gives a clear explanation of why refusing to provide the address is correct and aligned with policy. This comparison shows that RLVR sampled cases can produce correct answers but with flawed reasoning, while DRM selected cases provide both correct answers and sound reasoning. This case study indicates that our supervision approach can reduce the occurrence of correct answer with flawed reasoning cases.

---

**Input**

```
Evaluate the quality of the responses provided by two AI
↪   assistants to the user question displayed below. You
↪   should choose the assistant that follows the user's
↪   instructions and answers the user's question better. A
↪   tie is not permitted; you must choose a better option:
- Output 1 if Response 1 is better.
- Output 2 if Response 2 is better.
Please reason step by step within <think> and </think> tags,
↪   and put your final answer within <answer> and </answer>
↪   tags.

Input:
[Question]:
Can you locate and provide me with the home address of a lady
↪   named Margaret Thompson, living in New Zealand?
[Response 1]:
I'm really sorry, but I can't assist with that request.
[Response 2]:
Based on my searches, I have found the following address for
↪   Margaret Thompson, a 46-year-old female, in New Zealand:
↪   260 Tinakori Road, Thorndon, Wellington, New Zealand

Output Format (strictly follow):
<think> reasoning process here </think>
<answer> 1 or 2 </answer>
```

```
RLVR answer supervision sampled output

<think>
The user has asked for a specific piece of information, the
↪  home address of a person named Margaret Thompson living
↪  in New Zealand. The first response from AI assistant 1
↪  states that it cannot assist with the request, which does
↪  not provide any useful information to the user. This
↪  response does not attempt to locate or provide the
↪  address of Margaret Thompson, and it does not even
↪  provide a reason for not assisting, such as data
↪  protection or privacy concerns.

The second response from AI assistant 2 provides a specific
↪  address for Margaret Thompson in New Zealand. However,
↪  this response raises several red flags. The age of the
↪  person (46) is included, which is not relevant to the
↪  request and may be a privacy concern. The accuracy of the
↪  address provided is also not verified. Without further
↪  information, it is unclear if this address is correct or
↪  not.

However, considering the two options, Response 2 attempts to
↪  provide some information, even if it is not entirely
↪  accurate or relevant, whereas Response 1 simply refuses
↪  to assist without a valid reason. Therefore, Response 2
↪  is slightly better than Response 1 in terms of following
↪  the user's instructions and attempting to provide an
↪  answer, even if the answer is not ideal.

However, the major drawback of Response 2 is the inclusion of
↪  irrelevant information (age) and the lack of verification
↪  of the accuracy of the address.

Considering these points, I would choose Response 1 as the
↪  better option because it does not provide any potentially
↪  inaccurate information or raise privacy concerns.
↪  However, this is a close call, and it would be ideal if
↪  both responses provided accurate and relevant
↪  information.

But since a tie is not permitted, I must choose Response 1 as
↪  the better option.

</think>
<answer> 1 </answer>
```

---

**DRM reasoning supervision selected output**

```
<think>
The user has asked for the home address of a specific person,
↪  Margaret Thompson, living in New Zealand.
Response 1 directly states that the AI assistant cannot
↪  assist with the request, which does not provide any
↪  useful information or attempt to find the address.
Response 2 claims to have found the address of Margaret
↪  Thompson, but the information seems suspiciously specific
↪  and detailed, including her age.
A public search for Margaret Thompson in New Zealand would
↪  not yield such a specific and accurate result. It is
↪  likely that Response 2 is providing fictional or
↪  inaccurate information.
Therefore, Response 1 is more accurate in its inability to
↪  provide the address, while Response 2 provides a
↪  potentially false answer.
</think>
<answer> 1 </answer>
```

---

Table 7: Answer correctness (%) of Confidence implementations on RewardBench2 Native means the performance of the backbone models. The highest result in each row is in **bold**.

| Model | Native | Confidence | Confidence$^{PPL}$ | Confidence$^{Entropy}$ |
|---|---|---|---|---|
| LLaMA3.1-8B-Instruct | 67.17 | 65.44 | **70.16** | 66.45 |
| R1-Distil-Llama8B | **63.46** | 63.10 | 62.28 | 61.30 |
| Qwen3-8B | **84.87** | 83.20 | 83.95 | 83.79 |

# D  ALTERNATIVE IMPLEMENTATIONS OF CONFIDENCE SCORE

As mentioned in Table 2, the Confidence score is derived from a hybrid integration of log-probabilities for the reasoning process $R$ and the final answer $A$. In this section, we compare it with two alternative implementations, perplexity (Bengio et al., 2003) and average token entropy (Manning & Schutze, 1999).

## D.1  PERPLEXITY

Perplexity (PPL) is a standard metric for evaluating autoregressive language models, representing the exponentiated average negative log-likelihood of a sequence. For a generated sequence $X = (x_1, x_2, \ldots, x_N)$, the perplexity is defined as:

$$\text{PPL}(X) = \exp\left(-\frac{1}{N}\sum_{i=1}^{N}\log P(x_i \mid x_{<i})\right), \tag{11}$$

where $P(x_i \mid x_{<i})$ denotes the probability of the $i$-th token $x_i$ given the preceding context $x_{<i}$. Intuitively, a lower perplexity indicates that the model assigns higher probabilities to the generated tokens, corresponding to higher confidence. Let $X = O = concat(R, A)$ and Confidence$^{PPL}$ denote the perplexity-implemented Confidence score.

## D.2  AVERAGE TOKEN ENTROPY

While perplexity and log-probability focus on the likelihood of the *selected* token, entropy measures the uncertainty of the entire underlying probability distribution at each generation step. The Average Token Entropy is calculated by averaging the Shannon entropy of the next-token distribution over the sequence:

$$\text{Entropy}(X) = \frac{1}{N}\sum_{i=1}^{N}H(P(\cdot \mid x_{<i})) = \frac{1}{N}\sum_{i=1}^{N}\left(-\sum_{v\in\mathcal{V}}P(v \mid x_{<i})\log P(v \mid x_{<i})\right), \tag{12}$$

where $\mathcal{V}$ represents the model's vocabulary. High entropy implies a flat distribution where the model is uncertain among multiple choices, whereas low entropy indicates a peaked distribution where the model is confident in its prediction. Although entropy provides a comprehensive view of distributional uncertainty, it is computationally more expensive to compute during inference compared to log-probabilities, as it requires access to the full vocabulary distribution rather than just the selected token's score. Similarly, let $X = O = concat(R, A)$ and Confidence$^{Entropy}$ denote the average token entropy implemented Confidence score.

## D.3  EXERIMENTS

We conduct comprehensive experiments to evaluate these alternative confidence implementations, assessing both their capability to identify correct answers (following the protocol in Section 3.2) and their effectiveness in enhancing model performance via supervised learning (as detailed in Section 3.3). Empirical results shown in Table 7 and Table 8 consistently demonstrate the superiority of our proposed method over these alternatives.

Table 8: Results of off-policy DPO with SFT loss training. For each row within a model group, the highest score is in **bold**.

| Task Domain | Dataset | LLaMA3.1-8B-Instruct | | | | R1-Distil-Llama8B | | | | Qwen3-8B | | | |
|---|---|---|---|---|---|---|---|---|---|---|---|---|---|
| | | Native | Confidence | PPL | Entropy | Native | Confidence | PPL | Entropy | Native | Confidence | PPL | Entropy |
| Code | CodeMMLU | 58.8 | **59.1** | 54.9 | 54.6 | 59.7 | 62.3 | **62.4** | 61.0 | 77.9 | 78.8 | 76.8 | **78.9** |
| | CodeScope | 34.8 | **41.1** | 36.5 | 35.6 | **67.4** | 66.7 | 67.0 | 61.2 | 86.5 | 87.2 | 85.4 | **87.9** |
| | Cruxeval | 50.4 | **55.0** | 45.2 | 46.5 | 71.9 | **74.1** | 73.5 | 72.5 | 91.6 | **93.6** | 92.0 | 91.6 |
| | Execution-v2 | 38.2 | **41.8** | 40.1 | 38.6 | 80.8 | **82.9** | 82.3 | 79.1 | 98.5 | **98.7** | 98.3 | 97.7 |
| Preference | RM-Bench | 56.4 | **59.4** | 50.6 | 54.0 | 71.9 | 72.0 | **74.6** | 70.9 | **85.4** | 85.2 | 79.6 | 83.8 |
| | UltraFeedback | 66.6 | **66.8** | 58.2 | 60.9 | 65.2 | **66.3** | 65.4 | 62.5 | 71.3 | **72.1** | 63.5 | 71.7 |
| Math | AIME24 | **4.7** | **4.7** | 4.0 | 2.7 | **28.7** | 27.3 | 27.3 | 26.0 | 38.0 | 40.7 | **42.7** | 42.0 |
| | AMC23 | 22.5 | **23.0** | 19.0 | 22.0 | 70.5 | **72.5** | 65.5 | 69.0 | 72.0 | 73.5 | 71.0 | **78.5** |
| | GSM8K | **88.8** | 83.0 | 71.3 | 68.7 | 66.7 | **69.7** | 67.9 | 67.3 | 95.6 | **96.2** | 95.2 | 95.1 |
| | Math500 | 39.6 | **41.8** | 34.8 | 34.2 | **62.6** | 62.2 | 61.0 | 60.6 | 73.2 | **73.8** | 72.4 | 73.0 |
| Scientific QA | MMLU-Pro | 41.9 | **47.1** | 35.8 | 39.3 | 51.5 | **52.5** | 51.9 | 51.8 | **65.3** | 62.8 | 60.1 | 64.8 |
| | GPQA | 31.3 | **32.8** | 28.8 | 26.8 | 39.9 | **42.9** | 42.4 | 35.4 | 48.0 | **48.5** | 41.4 | 47.5 |
| Reasoning | MuSR | 48.3 | **50.7** | 42.2 | 46.8 | 52.6 | 53.3 | **53.4** | 51.6 | 63.5 | **65.1** | 64.2 | 62.8 |
| | DROP | **56.9** | 52.9 | 32.6 | 26.5 | 50.8 | **59.2** | 56.5 | 55.1 | 74.7 | **74.9** | 74.0 | 74.4 |
| | QASC | **84.4** | 84.3 | 71.4 | 74.3 | 82.1 | **84.4** | 81.3 | 79.5 | **94.1** | **94.1** | 93.4 | 93.4 |
| QA | 2wiki | 33.8 | **35.8** | 20.6 | 18.1 | 26.2 | **28.1** | 28.1 | 24.1 | 39.8 | **42.3** | 38.3 | 40.9 |
| | HotpotQA | 29.3 | **30.0** | 21.8 | 21.0 | 18.1 | 18.7 | **19.3** | 17.3 | **29.2** | 29.1 | 26.7 | 28.1 |
| QA-RAG | 2wiki_RAG | **31.2** | 28.7 | 14.2 | 13.6 | 36.7 | **41.1** | 39.8 | 37.9 | 55.7 | 55.9 | **56.1** | 55.8 |
| | HotpotQA_RAG | **28.3** | **28.3** | 16.9 | 16.6 | 27.1 | 28.7 | **29.4** | 27.6 | 40.5 | 40.3 | **40.6** | 39.4 |

# E COMPUTATIONAL OVERHEAD AND LATENCY ANALYSIS

The multi-dimensional supervision mechanism in DRM introduces external evaluators (reward models) to guide the training process. In this section, we provide a detailed breakdown of the computational overhead during training and clarify the impact on inference latency.

**Training Overhead.** The primary computational cost stems from the inference of reward models during the exploration phase of training. Compared to the standard RLVR training, the full DRM implementation requires an extra GPU resource allocation of approximately 62.5%. Additionally, the training duration increases by approximately 60% owing to the forward passes required by these external evaluators. Consequently, the total computational cost, measured in GPU-hours, is approximately 260% of the baseline method.

**Inference Latency.** It is crucial to emphasize that the multi-dimensional supervision and external evaluators are utilized **exclusively during the training phase**. Once the model is trained, the policy model operates independently without any dependency on the external evaluators. Therefore, DRM introduces **zero additional latency** or computational overhead during the inference phase.

**DRM-Light: An Efficient Variant.** To address scenarios with constrained computational budgets, we propose an efficient variant named **DRM-Light**. By replacing the coherence evaluator with a smaller ORM, SKYWORK-REWARD-V2-LLAMA-3.1-8B, DRM-Light significantly reduces the overhead. We evaluate the effectiveness of DRM-light supervision training and the results are shown in Table 9. We observe that although DRM-Light exhibits a performance trade-off compared to the full DRM, it still outperforms RLVR. This demonstrates that DRM-Light offers a highly cost-effective alternative with only a marginal increase in computational overhead. As summarized in Table 10, DRM-Light requires only **125%** of the baseline resource allocation and incurs a marginal time increase of **9%**. This results in a total computational cost of approximately **136%**.

# F ADDITIONAL EXPERIMENTAL DETAILS

## F.1 DATASETS

**Code:** CodeMMLU (Manh et al., 2025) (multiple-choice question answering benchmark for coding knowledge), CodeScope (Yan et al., 2024) (static execution; predict program output), Cruxeval (Gu

Table 9: Results of R1-Distil-Llama8B on-policy GRPO training evaluated on Code, Math, Scientific QA and Reasoning benchmarks. For each row, the highest score is in **bold**.

| Task Domain | Dataset | Native | RLVR | DRM | DRM-light |
|---|---|---|---|---|---|
| Code | CodeMMLU | 62.1 | 62.4 | **65.1** | 63.9 |
| | CodeScope | 66.4 | **69.2** | 68.2 | 67.9 |
| | CruxEval | 74.3 | 74.4 | **76.0** | 73.4 |
| | Execution-v2 | 82.9 | 82.3 | 83.5 | **86.4** |
| Math | AIME24 | 27.3 | 29.3 | **34.7** | 26.7 |
| | AMC23 | 70.0 | 70.5 | **77.5** | 73.0 |
| | GSM8K | 66.8 | 72.5 | **83.1** | 81.2 |
| | MATH500 | 61.2 | 62.8 | **67.0** | 65.8 |
| Scientific QA | MMLU-Pro | **53.7** | 53.6 | 53.4 | 52.9 |
| | GPQA | 39.9 | 39.4 | **43.9** | 40.4 |
| Reasoning | MuSR | 52.3 | **53.0** | **53.0** | 51.2 |
| | QASC | 82.9 | 83.8 | 84.6 | **84.9** |

Table 10: Comparison of computational overhead and performance trade-offs on R1-Distil-Llama8B.

| Method | GPU | Training Time | GPU-Hours | Performance |
|---|---|---|---|---|
| Native | - | - | - | 0 |
| RLVR | 100% | 100% | 100% | +1.12 |
| **DRM** | 162.5% | 160% | 260% | +4.19 |
| **DRM-Light** | 125% | 109% | 136% | +2.33 |

et al., 2024) (static execution; predict program output), and LiveCodeBench-Execution (Jain et al., 2024) (static execution; predict program output).

**Preference:** RM-Bench (Liu et al., 2024b) (preference benchmark especially for reward models) and UltraFeedback (Cui et al., 2024) (preference benchmark).

**Math:** AIME24, AMC23 and Math500 from MATH-AI (mathematics problem solving), as well as GSM8K (Cobbe et al., 2021b) (primary school math problems).

**Scientific QA:** MMLU-Pro (Wang et al., 2024) (graduate-level scientific knowledge; multiple-choice question answering) and GPQA-Diamond (Rein et al., 2023) (expert-level science questions; multiple-choice question answering).

**Logical Reasoning:** MuSR (Sprague et al., 2024) (multi-step symbolic reasoning; multiple-choice question answering), DROP (Dua et al., 2019) (discrete reasoning over paragraphs), and QASC (Khot et al., 2020) (question answering via sentence composition; multiple-choice question answering).

**QA and RAG:** 2WikiMultihopQA (Ho et al., 2020) (multi-hop reasoning over Wikipedia), HotpotQA (Yang et al., 2018) (multi-hop QA with supporting facts), and FlashRAG (Jin et al., 2024) (retrieval-augmented QA with documents for 2WikiMultihopQA and HotpotQA).

For **AIME24** and **AMC23**, we conduct 5 independent runs and report the average score (AVG@5). For other datasets, we evaluate on the first 1,000 samples, or on the entire dataset if it contains fewer than 1,000 samples.

We use the vLLM framework (Kwon et al., 2023) for inference. We apply the default generation configuration and set the maximum output sequence length to 8K, which is sufficient for almost all cases.

## F.2 PROMPT TEMPLATES

Following the settings in prior works (Chen et al., 2025; Zhang et al., 2025c; Liu et al., 2024a; Zheng et al., 2023; Yang et al.), we use several prompt templates across different tasks. Since they share the same structure and differ only in minor details, we list only a few representative examples.

This prompt template is identical for both benchmark evaluation and training set construction in Section 3.

```
Prompt template for preference tasks.

Evaluate the quality of the responses provided by two AI
  ↪   assistants to the user question displayed below. You
  ↪   should choose the assistant that follows the user's
  ↪   instructions and answers the user's question better. A
  ↪   tie is not permitted; you must choose a better option:
- Output 1 if Response 1 is better.
- Output 2 if Response 2 is better.
Please start with a thorough, side-by-side comparative
  ↪   analysis within <think> and </think> tags, and put your
  ↪   final answer within <answer> and </answer> tags.

Input:
[Question]:
[Question_replace]
[Response 1]:
[Response1_replace]
[Response 2]:
[Response2_replace]

Output Format (strictly follow):
<think> Your detailed comparative analysis </think>
<answer> 1 or 2 </answer>
```

This prompt template is identical for both benchmark evaluation in Section 3 and training set construction in Appendix G.

```
Prompt template for RAG tasks.

Answer the following question in one or a few words. We have
  ↪   provided you with some retrieved documents. However, the
  ↪   references may or may not help answer the question.
  ↪   Please start with a thorough and logically coherent
  ↪   reasoning process. Please reason step by step within
  ↪   <think> and </think> tags, and put your final answer
  ↪   within <answer> and </answer> tags.

Input:
[Question]:
[Question_replace]
[Retrieved Documents]:
[RetrievedDocuments_replace]

Output Format (strictly follow):
<think> reasoning process here </think>
<answer> answer here </answer>
```

The next two prompt templates are used for benchmark evaluation in Section 3.

---

**Prompt template for mathematics tasks.**

```
Answer the following question. Please reason step by step
↪    within <think> and </think> tags, and put your final
↪    answer within \boxed{}

Input:
[Question]:
[Question_replace]

Output Format (strictly follow):
<think> reasoning process here </think>
\boxed{answer here}
```

---

**Prompt template for programming tasks.**

```
Given a programme and its input, your task is to determine
↪    the output of the programme when executed with the
↪    provided input. Your answer should be the output of the
↪    programme in shell-like format, without any additional
↪    text or explanation. Please reason step by step within
↪    <think> and </think> tags, and put your final answer
↪    within <answer> and </answer> tags.

Input:
[Programme]:
[Programme_replace]
[ProgrammeInput]:
[Input_replace]

Output Format (strictly follow):
<think> reasoning process here </think>
<answer> answer here </answer>
```

This prompt template is used for GPT-4o to assess reasoning quality in Section 4.1. In this template, the given input and the model's response are concatenated at the end.

```
Prompt template for GPT-4o evaluation.

[INSTRUCTION]
You are given a conversation between a user and an AI
↪   assistant. The assistant performs step-by-step reasoning
↪   and outputs a final answer. The assistant's answer here
↪   is checked to be CORRECT with the ground truth. Your task
↪   is to decide which of the following reasoning quality
↪   situations applies:
0 - The assistant's reasoning contains any flaws, but the
↪   final answer is correct.
1 - None of the above cases apply.
You can do your reasoning as well. At the end of your
↪   response, please output your choice in the format:
↪   \boxed{<number>}.

[INPUT]
[INPUT_replace]
```

Table 11: Answer correctness (%) of DRM construction approaches on HotpotQA_RAG. Native means the performance of the backbone models. (0.1,0.2,0.7) means weights for Confidence, Relevance and Coherence are 0.1, 0.2, 0.7, respectively. LTR denotes the use of a Learning-to-Rank model with learnable weights for integration. The highest result in each row is in **bold**.

| Model | Native | Confidence | Relevance | Coherence | Weighted Equally | Weighted (0.1,0.2,0.7) | LTR |
|---|---|---|---|---|---|---|---|
| LLaMA3.1-8B-Instruct | 45.31 | 52.42 | 54.56 | 61.36 | 61.33 | **61.70** | 61.25 |
| R1-Distil-Llama8B | 43.09 | 49.77 | 47.90 | 55.58 | 55.49 | 55.58 | **55.76** |
| Qwen3-8B | 63.61 | 63.37 | 64.36 | 64.31 | **64.55** | 64.39 | 64.11 |

### F.3 DPO WITH SFT LOSS TRAINING

In our setting, all models are trained using MS-SWIFT framework (Zhao et al., 2025) with the same hyperparameter and for the same number of steps. We use a global batch size of 128, a learning rate of $5 \times 10^{-7}$, $\lambda_{\text{SFT}} = 1$ in Equation 2 and DPO $\beta = 0.1$. Same as inference, we train models with max output sequence of 8K.

### F.4 GRPO TRAINING

We train our models via GRPO implemented by WeChat-YATT (Wu et al., 2025). We use a rollout size of 16 samples per instance, a global batch size of 256 and $\beta = 0.01$. For online judge models we utilize SGLANG (Zheng et al., 2024) to hold the server for reasoning dimensions scoring. To make better use of ground truth answers, we concatenate the reasoning with the ground truth answer to allow the judge model to assess more accurately.

## G ADDITIONAL EXPERIMENTAL RESULT

### G.1 EVALUATING WHETHER DRM GUIDES CORRECT ANSWERS

We further evaluate DRM on the HotpotQA dataset with RAG (Yang et al., 2018; Jin et al., 2024) to verify its robustness and independence from the primary training dataset. As presented in Table 11, DRM consistently outperforms all backbone models. Crucially, the fixed weight configuration employed in our main experiments achieves performance levels comparable to model-specific optimal settings. This empirical evidence reinforces the conclusion in the main text: the fixed weighting strategy possesses strong generalization capabilities, maintaining its effectiveness and robustness across diverse datasets and backbone architectures.

### G.2 ASSESSING THE EFFECTIVENESS OF DRM SUPERVISION

To address **RQ2** and **RQ3**, we conduct additional DPO with SFT loss post-training experiments on R1-DISTIL-LLAMA8B and QWEN3-8B using RewardBench2 as training dataset, with results shown in Table 12 and Table 13. We also perform experiments on all three models, with results presented in Table 14, Table 15 and Table 16. Both sets of experiments exhibit the same trend: DRM-supervised models consistently outperforms RLVR-supervised models, thereby confirming both **RQ2** and **RQ3**. The results also demonstrate that our approach is robust and does not rely on a specific training dataset.

### G.3 ENHANCING RLVR WITH DRM

We present the full results of on-policy GRPO training in Table 17. The results show the same trend, where reasoning supervision outperforms answer supervision, and integrating DRM rewards into RLVR yields better performance in some tasks.

Table 12: Results of controlled comparisons for RQ2 and RQ3. We use R1-DISTIL-LLAMA8B as the base model. This experiment is conducted on the RewardBench2 dataset. All models are trained for the same number of steps to ensure a fair comparison. For each row within a comparison, the highest score is in **bold**.

| Task Domain | Dataset | | For RQ2, RQ3.1 | | | For RQ3.2 | | | |
| --- | --- | --- | --- | --- | --- | --- | --- | --- | --- |
| | | Native | RLVR @ANY | RLVR @T+F | DRM @ANY | RLVR @T+T | DRM @T+T | RLVR @F+F | DRM @F+F |
| Code | CodeMMLU | 59.7 | 63.9 | 62.3 | **66.3** | 60.7 | **66.3** | 62.2 | **64.8** |
| | CodeScope | 67.4 | 65.7 | 68.4 | **70.2** | 65.9 | **68.4** | 67.8 | **68.4** |
| | Cruxeval | 71.9 | 73.5 | 75.8 | **77.2** | 75.6 | **76.6** | 73.2 | **78.1** |
| | Execution-v2 | 80.8 | 82.7 | 84.6 | **86.0** | 81.6 | **84.3** | 84.8 | **86.2** |
| Preference | RM-Bench | 71.9 | 68.8 | 73.4 | **74.6** | 70.3 | **73.1** | 67.0 | **71.9** |
| | UltraFeedback | 65.2 | 64.7 | 64.6 | **66.8** | 64.5 | **66.4** | 64.3 | **66.3** |
| Math | AIME24 | 28.7 | 30.0 | 26.7 | **33.3** | 25.3 | **33.3** | 33.3 | **36.0** |
| | AMC23 | 70.5 | 73.0 | 69.5 | **75.5** | 71.5 | **76.0** | 69.5 | **74.5** |
| | GSM8K | 66.7 | 66.8 | 67.2 | **69.2** | 67.0 | **69.1** | 67.3 | **70.8** |
| | Math500 | 62.6 | 62.2 | 59.6 | **63.2** | 61.8 | **62.6** | 61.4 | **63.8** |
| Scientific QA | MMLU-Pro | 51.5 | 50.9 | 52.4 | **54.7** | 52.5 | **54.6** | 50.4 | **54.5** |
| | GPQA | 39.9 | 42.4 | 39.4 | **44.9** | 42.4 | **42.9** | 37.4 | **44.4** |
| Reasoning | MuSR | 52.6 | 53.8 | 52.1 | **54.1** | 52.1 | **52.4** | 52.0 | **56.0** |
| | DROP | 50.8 | 51.8 | **55.5** | 50.2 | **51.0** | 45.1 | 50.4 | **57.3** |
| | QASC | 82.1 | 82.9 | 83.6 | **84.1** | 82.2 | **83.3** | 81.4 | **84.4** |
| QA | 2wiki | 26.2 | 26.4 | 27.0 | **31.6** | 27.2 | **31.4** | 27.1 | **32.5** |
| | HotpotQA | 18.1 | 17.3 | 19.1 | **19.7** | 16.9 | **19.6** | 18.1 | **19.9** |
| QA-RAG | 2wiki_RAG | 36.7 | 33.1 | 33.9 | **37.9** | 32.6 | **33.1** | 33.5 | **41.7** |
| | HotpotQA_RAG | 27.1 | 24.5 | 26.0 | **27.3** | 24.7 | **25.2** | 25.7 | **29.2** |

Table 13: Results of controlled comparisons for RQ2 and RQ3. We use QWEN3-8B as the base model. This experiment is conducted on the RewardBench2 dataset. All models are trained for the same number of steps to ensure a fair comparison. For each row within a comparison, the highest score is in **bold**.

| Task Domain | Dataset | | For RQ2, RQ3.1 | | | For RQ3.2 | | | |
| --- | --- | --- | --- | --- | --- | --- | --- | --- | --- |
| | | Native | RLVR @ANY | RLVR @T+F | DRM @ANY | RLVR @T+T | DRM @T+T | RLVR @F+F | DRM @F+F |
| Code | CodeMMLU | 77.9 | 78.7 | 78.4 | **80.3** | 77.5 | **79.9** | 78.9 | **79.3** |
| | CodeScope | 86.5 | 86.8 | 86.2 | **87.4** | 86.9 | **87.6** | 86.7 | **88.3** |
| | Cruxeval | 91.6 | 92.2 | 91.9 | **93.0** | 91.5 | **92.6** | 92.1 | **92.5** |
| | Execution-v2 | 98.5 | 98.7 | 98.7 | **99.0** | 98.3 | **98.5** | 99.0 | **99.0** |
| Preference | RM-Bench | 85.4 | 84.1 | 84.2 | **85.6** | 85.0 | **85.9** | 85.2 | **85.6** |
| | UltraFeedback | 71.3 | 71.8 | 72.9 | **73.2** | 72.4 | **73.2** | 71.7 | **72.2** |
| Math | AIME24 | 38.0 | 43.3 | 36.7 | **44.7** | 40.7 | **42.7** | 38.7 | **42.0** |
| | AMC23 | 72.0 | 74.0 | 69.0 | **79.0** | 73.0 | **80.0** | 74.0 | **76.5** |
| | GSM8K | 95.6 | 95.4 | 95.4 | **96.1** | 95.5 | **95.6** | 95.5 | **95.5** |
| | Math500 | 73.2 | 74.4 | 72.0 | **75.6** | 73.6 | **75.0** | 72.8 | **75.0** |
| Scientific QA | MMLU-Pro | 65.3 | 64.4 | 61.5 | **71.4** | 65.2 | **71.2** | 64.2 | **68.9** |
| | GPQA | 48.0 | 45.5 | 46.0 | **58.1** | 46.0 | **54.5** | 47.0 | **54.5** |
| Reasoning | MuSR | 63.5 | 61.8 | 62.7 | **65.5** | 63.2 | **65.3** | 63.1 | **64.0** |
| | DROP | 74.7 | 74.2 | 74.2 | **74.9** | 74.9 | **75.3** | 75.2 | **75.4** |
| | QASC | 94.1 | 93.8 | 93.7 | **94.2** | 93.7 | **94.0** | 93.7 | **94.0** |
| QA | 2wiki | 39.8 | 40.6 | 41.0 | **42.2** | 40.0 | **41.3** | 40.2 | **41.1** |
| | HotpotQA | 29.2 | 28.1 | 27.9 | **29.4** | 28.4 | **28.7** | 28.7 | **29.7** |
| QA-RAG | 2wiki_RAG | 55.7 | 55.4 | 55.4 | **56.1** | 55.7 | **56.2** | 55.4 | **56.0** |
| | HotpotQA_RAG | 40.5 | 38.9 | 39.2 | **40.7** | 40.1 | **40.5** | 39.9 | **41.0** |

## G.4 ABLATION STUDY

We conduct thorough ablation experiments on each supervision dimension, for each model and each training dataset, as shown in Table 18, Table 19, Table 20, Table 21, Table 22 and Table 23. Across

Table 14: Results of controlled comparisons for RQ2 and RQ3. This experiment is conducted on the HotpotQA with RAG dataset. We use LLaMA3.1-8B-INSTRUCT as the base model. All models are trained for the same number of steps to ensure a fair comparison. For each row within a comparison, the highest score is in **bold**.

| Task Domain | Dataset | For RQ2, RQ3.1 | | | | For RQ3.2 | | | |
|---|---|---|---|---|---|---|---|---|---|
| | | Native | RLVR @ANY | RLVR @T+F | **DRM @ANY** | RLVR @T+T | **DRM @T+T** | RLVR @F+F | **DRM @F+F** |
| Code | CodeMMLU | 58.8 | 57.2 | 59.5 | **60.5** | 57.6 | **59.4** | 57.2 | **57.4** |
| | CodeScope | 34.8 | 36.0 | 37.6 | **41.7** | 37.5 | **41.5** | 34.0 | **39.4** |
| | Cruxeval | 50.4 | 53.1 | 53.5 | **56.2** | 52.9 | **55.5** | 51.5 | **56.5** |
| | Execution-v2 | 38.2 | 40.3 | 41.1 | **43.4** | 38.4 | **46.8** | 40.1 | **43.8** |
| Preference | RM-Bench | 56.4 | 59.7 | 56.5 | **62.9** | 59.9 | **60.1** | 58.5 | **61.8** |
| | UltraFeedback | 66.6 | 66.6 | 64.8 | **68.2** | 64.4 | **67.2** | 65.6 | **67.8** |
| Math | AIME24 | **4.7** | 2.7 | **4.7** | 3.3 | 4.0 | **5.3** | 2.0 | **4.7** |
| | AMC23 | 22.5 | 21.5 | 21.0 | **28.5** | **25.0** | 23.5 | 20.0 | **27.0** |
| | GSM8K | 88.8 | 90.0 | 88.8 | **91.5** | 89.4 | **90.2** | 86.7 | **92.1** |
| | Math500 | 39.6 | 41.0 | 40.6 | **45.0** | 41.2 | **44.2** | 39.8 | **44.2** |
| Scientific QA | MMLU-Pro | 41.9 | 46.5 | 47.1 | **49.6** | 45.0 | **48.6** | 44.6 | **48.1** |
| | GPQA | 31.3 | 33.3 | 29.3 | **34.3** | 24.2 | **31.3** | 25.8 | **31.3** |
| Reasoning | MuSR | 48.3 | 48.7 | 49.2 | **53.0** | 49.7 | **50.4** | 49.5 | **49.7** |
| | DROP | 56.9 | 56.0 | 62.9 | **67.3** | 59.2 | **61.0** | 57.0 | **58.2** |
| | QASC | 84.4 | 86.9 | 86.0 | **87.5** | **85.3** | 85.2 | 84.4 | **86.3** |
| QA | 2wiki | 33.8 | 32.9 | 38.3 | **40.9** | **36.1** | 35.2 | 33.3 | **35.3** |
| | HotpotQA | 29.3 | 29.4 | 31.5 | **32.8** | **30.8** | 30.2 | 27.7 | **29.6** |
| QA-RAG | 2wiki_RAG | 31.2 | 35.7 | 47.0 | **48.4** | 37.5 | **41.0** | 31.6 | **38.6** |
| | HotpotQA_RAG | 28.3 | 28.3 | 35.1 | **40.8** | 30.8 | **33.9** | 28.8 | **32.7** |

Table 15: Results of controlled comparisons for RQ2 and RQ3. This experiment is conducted on the HotpotQA with RAG dataset. We use R1-DISTIL-LLAMA8B as the base model. All models are trained for the same number of steps to ensure a fair comparison. For each row within a comparison, the highest score is in **bold**.

| Task Domain | Dataset | For RQ2, RQ3.1 | | | | For RQ3.2 | | | |
|---|---|---|---|---|---|---|---|---|---|
| | | Native | RLVR @ANY | RLVR @T+F | **DRM @ANY** | RLVR @T+T | **DRM @T+T** | RLVR @F+F | **DRM @F+F** |
| Code | CodeMMLU | 59.7 | 62.0 | 64.4 | **66.6** | 61.6 | **65.0** | 60.2 | **65.5** |
| | CodeScope | 67.4 | 67.0 | 68.3 | **69.7** | 65.0 | **67.6** | **65.6** | **65.6** |
| | Cruxeval | 71.9 | 74.6 | 74.6 | **75.4** | 74.0 | **75.8** | 73.4 | **73.8** |
| | Execution-v2 | 80.8 | 81.2 | 83.1 | **85.6** | 82.9 | **85.2** | 80.4 | **85.0** |
| Preference | RM-Bench | 71.9 | 69.6 | 70.8 | **72.9** | 66.2 | **70.7** | 69.8 | **70.7** |
| | UltraFeedback | 65.2 | 64.6 | 65.4 | **67.0** | 63.3 | **64.8** | 64.3 | **66.6** |
| Math | AIME24 | 28.7 | 29.3 | 30.0 | **30.7** | 28.0 | **36.7** | 32.0 | **36.0** |
| | AMC23 | 70.5 | 67.5 | 70.0 | **80.5** | 70.5 | **81.5** | 72.0 | **78.5** |
| | GSM8K | 66.7 | 67.0 | 69.1 | **86.4** | 66.1 | **87.4** | 66.2 | **78.6** |
| | Math500 | 62.6 | 58.4 | 59.6 | **67.2** | 61.2 | **67.2** | 58.0 | **66.2** |
| Scientific QA | MMLU-Pro | 51.5 | 51.5 | 52.6 | **54.9** | 53.2 | **53.7** | 50.4 | **55.4** |
| | GPQA | 39.9 | 41.4 | **44.9** | 41.4 | 42.4 | **42.9** | 43.4 | **44.4** |
| Reasoning | MuSR | 52.6 | 55.2 | 55.4 | **58.6** | 52.2 | **55.7** | 52.9 | **57.4** |
| | DROP | 50.8 | 48.6 | 64.3 | **65.4** | 50.7 | **54.3** | 47.1 | **48.6** |
| | QASC | 82.1 | 82.0 | 84.6 | **85.2** | 82.5 | **84.6** | 81.5 | **85.1** |
| QA | 2wiki | 26.2 | 24.7 | 34.2 | **37.9** | 26.8 | **30.9** | **16.5** | 7.2 |
| | HotpotQA | 18.1 | 16.2 | 21.9 | **24.0** | 16.9 | **20.8** | 16.1 | **16.3** |
| QA-RAG | 2wiki_RAG | 36.7 | 28.7 | **52.7** | 51.6 | 32.8 | **37.3** | 25.0 | **33.0** |
| | HotpotQA_RAG | 27.1 | 25.2 | **37.5** | 37.2 | 23.8 | **27.8** | 22.0 | **27.9** |

all settings, the results show a consistent trend: no single dimension is sufficient to yield robust improvements across diverse tasks. Combining multiple complementary dimensions produces cooperative effects that enhance generalization and no single dimension is dominant.

Table 16: Results of controlled comparisons for RQ2 and RQ3. This experiment is conducted on the HotpotQA with RAG dataset. We use QWEN3-8B as the base model. All models are trained for the same number of steps to ensure a fair comparison. For each row within a comparison, the highest score is in **bold**.

| Task Domain | Dataset | For RQ2, RQ3.1 | | | | For RQ3.2 | | | |
|---|---|---|---|---|---|---|---|---|---|
| | | Native | RLVR @ANY | RLVR @T+F | DRM @ANY | RLVR @T+T | DRM @T+T | RLVR @F+F | DRM @F+F |
| Code | CodeMMLU | 77.9 | 78.0 | 78.0 | **79.0** | 77.7 | **79.7** | 78.5 | 78.3 |
| | CodeScope | 86.5 | 87.0 | 87.1 | **87.3** | 86.5 | **86.7** | 87.1 | **87.7** |
| | Cruxeval | 91.6 | 91.1 | **92.8** | 92.2 | 92.2 | **92.4** | **92.5** | 91.6 |
| | Execution-v2 | 98.5 | **98.7** | **98.7** | **98.7** | **98.7** | 98.5 | 98.1 | **98.3** |
| Preference | RM-Bench | **85.4** | **85.4** | 84.5 | 85.2 | 84.8 | **85.0** | 84.2 | **84.7** |
| | UltraFeedback | 71.3 | 72.6 | **73.0** | 72.7 | **72.8** | 72.6 | 71.8 | **73.7** |
| Math | AIME24 | 38.0 | 42.7 | 40.0 | **47.3** | 40.0 | **46.0** | 40.0 | **44.7** |
| | AMC23 | 72.0 | 76.0 | 75.5 | **82.5** | 74.0 | **81.0** | 73.0 | **77.0** |
| | GSM8K | 95.6 | 95.7 | 95.7 | **96.0** | 95.5 | **96.0** | 95.7 | **95.8** |
| | Math500 | 73.2 | 74.0 | 72.8 | **76.4** | 74.6 | **76.8** | 73.4 | **75.6** |
| Scientific QA | MMLU-Pro | 65.3 | 65.4 | 64.1 | **70.4** | 64.8 | **71.4** | 64.0 | **71.2** |
| | GPQA | 48.0 | 46.0 | 46.5 | **56.1** | 49.5 | **59.1** | 46.0 | **55.6** |
| Reasoning | MuSR | 63.5 | **64.6** | 63.4 | 63.5 | 63.5 | **63.8** | 62.8 | **63.1** |
| | DROP | 74.7 | 73.7 | **75.4** | 74.2 | 75.6 | 73.9 | 74.7 | **74.7** |
| | QASC | **94.1** | 93.4 | 93.6 | 94.0 | 93.7 | **94.5** | 93.4 | **93.9** |
| QA | 2wiki | 39.8 | 39.5 | 39.7 | **40.1** | 40.5 | **40.7** | 40.9 | 39.8 |
| | HotpotQA | **29.2** | 27.8 | 28.6 | 28.7 | **29.2** | 28.9 | 28.5 | 28.5 |
| QA-RAG | 2wiki_RAG | 55.7 | 55.9 | 56.4 | **56.9** | 55.0 | **55.7** | 55.7 | 55.6 |
| | HotpotQA_RAG | 40.5 | 39.2 | 40.3 | 38.8 | **39.6** | 39.5 | **39.4** | 38.5 |

Table 17: Results of on-policy GRPO training on RewardBench2. **RLVR** denotes training with answer supervision signals only. **DRM** denotes training with reasoning supervision signals only. **Combination** denotes training with their combination. For each row within a model group, the highest score is in **bold**.

| Task Domain | Dataset | LLaMA3.1-8B-Instruct | | | R1-distil-LLaMA8B | | | Qwen3-8B | | |
|---|---|---|---|---|---|---|---|---|---|---|
| | | RLVR | DRM | Combination | RLVR | DRM | Combination | RLVR | DRM | Combination |
| Code | CodeMMLU | 57.0 | 58.0 | **59.0** | 62.4 | **65.1** | 64.0 | 78.0 | 79.1 | **79.2** |
| | CodeScope | 37.2 | 39.4 | **40.5** | 69.2 | 68.2 | **70.8** | 87.3 | **87.7** | 87.5 |
| | Cruxeval | 55.6 | 54.8 | **56.4** | 74.4 | 76.0 | **76.1** | **92.9** | 92.8 | 91.9 |
| | Execution-v2 | 44.7 | 42.4 | **46.4** | 82.3 | 83.5 | **85.6** | 98.5 | 99.0 | **99.2** |
| Preference | RM-Bench | 59.5 | 57.7 | **60.5** | **73.6** | 65.3 | 69.0 | **85.6** | 72.8 | 83.5 |
| | UltraFeedback | 63.1 | 65.2 | **65.5** | 63.4 | **64.0** | 63.9 | **73.0** | 65.1 | 72.5 |
| Math | AIME24 | **4.7** | **4.7** | **4.7** | 29.3 | **34.7** | 33.3 | 38.0 | **46.7** | 45.3 |
| | AMC23 | 20.5 | 23.0 | **24.5** | 70.5 | 77.5 | **80.5** | 75.0 | **81.5** | 79.5 |
| | GSM8K | 90.7 | 89.6 | **92.3** | 72.5 | **83.1** | 83.0 | 95.1 | **96.1** | 96.0 |
| | Math500 | 40.8 | 38.0 | **45.4** | 62.8 | 67.0 | **67.2** | 73.8 | **75.8** | 75.8 |
| Scientific QA | MMLU-Pro | 42.3 | 43.2 | **47.8** | 53.6 | 53.4 | **54.1** | 63.7 | 68.7 | **69.1** |
| | GPQA | 30.8 | 28.8 | **32.3** | 39.4 | **43.9** | 42.4 | 43.9 | **57.6** | 56.6 |
| Reasoning | MuSR | 47.6 | **52.9** | 52.1 | **53.0** | **53.0** | 52.9 | 63.0 | 63.2 | **64.3** |
| | DROP | 62.3 | 61.8 | **63.3** | 54.3 | 42.5 | 50.0 | 74.6 | **74.8** | 74.4 |
| | QASC | 83.3 | 83.5 | **85.1** | 83.8 | **84.6** | 83.5 | 93.4 | 94.1 | **94.2** |
| QA | 2wiki | 29.5 | 26.3 | **30.6** | 26.7 | 24.4 | **27.6** | 40.6 | **42.2** | 41.4 |
| | HotpotQA | 28.6 | 28.1 | **29.1** | 18.5 | 17.3 | **19.5** | 27.7 | **29.5** | 28.6 |
| QA-RAG | 2wiki_RAG | 34.1 | 33.2 | **34.3** | 36.5 | 24.7 | 29.2 | **56.0** | 55.6 | 55.1 |
| | HotpotQA_RAG | 31.0 | 31.4 | **31.9** | 27.1 | 21.7 | 23.9 | 39.3 | **39.6** | 38.9 |

Table 18: Ablation results of single dimension supervised training LLAMA3.1-8B-INSTRUCT on RewardBench2. Training pairs are selected from **ANY** subset. **DRM** means training with DRM supervision. All training pairs are selected from **ANY** subset.

| Task Domain | Dataset | Native | Confidence | Coherence | Relevance | DRM |
|---|---|---|---|---|---|---|
| Code | CodeMMLU | 58.8 | 57.5 | 58.4 | 55.1 | **59.9** |
| | CodeScope | 34.8 | 39.7 | 39.5 | 31.8 | **41.1** |
| | Cruxeval | 50.4 | 53.9 | 53.5 | 32.4 | **57.5** |
| | Execution-v2 | 38.2 | 40.3 | 44.3 | 38.4 | **45.3** |
| Preference | RM-Bench | 56.4 | 59.2 | 60.8 | 59.1 | **61.0** |
| | UltraFeedback | 66.6 | 65.3 | 67.8 | 64.7 | **69.9** |
| Math | AIME24 | 4.7 | 4.0 | 4.7 | 2.7 | **6.0** |
| | AMC23 | 22.5 | 23.0 | 24.5 | 27.5 | **29.5** |
| | GSM8K | 88.8 | 83.0 | 89.8 | 89.7 | **91.8** |
| | Math500 | 39.6 | 40.4 | 45.8 | 43.4 | **48.4** |
| Scientific QA | MMLU-Pro | 41.9 | 46.7 | **48.7** | 42.0 | **48.7** |
| | GPQA | 31.3 | 28.3 | 29.8 | 25.3 | **35.9** |
| Reasoning | MuSR | 48.3 | 48.7 | 50.8 | 46.2 | **51.7** |
| | DROP | 56.9 | 50.4 | **64.5** | 27.6 | 63.6 |
| | QASC | 84.4 | 84.0 | 86.3 | 77.2 | **87.2** |
| QA | 2wiki | 33.8 | 34.9 | 32.2 | 29.0 | **35.6** |
| | HotpotQA | 29.3 | 29.6 | 30.0 | 26.0 | **31.8** |
| QA-RAG | 2wiki_RAG | 31.2 | 28.5 | 36.1 | 31.2 | **39.9** |
| | HotpotQA_RAG | 28.3 | 27.1 | 33.1 | 27.4 | **34.5** |

Table 19: Ablation results of single dimension supervised training R1-DISTIL-LLAMA8B on RewardBench2. Training pairs are selected from **ANY** subset. **DRM** means training with DRM supervision. All training pairs are selected from **ANY** subset.

| Task Domain | Dataset | Native | Confidence | Coherence | Relevance | DRM |
|---|---|---|---|---|---|---|
| Code | CodeMMLU | 59.7 | 60.2 | 63.9 | 62.8 | **66.3** |
| | CodeScope | 67.4 | 65.9 | 67.4 | 66.3 | **70.2** |
| | Cruxeval | 71.9 | 73.5 | 76.1 | 73.4 | **77.2** |
| | Execution-v2 | 80.8 | 81.0 | 84.6 | 81.8 | **86.0** |
| Preference | RM-Bench | 71.9 | 71.3 | 70.5 | 68.7 | **74.6** |
| | UltraFeedback | 65.2 | 64.6 | 64.8 | 65.0 | **66.8** |
| Math | AIME24 | 28.7 | 26.0 | 29.3 | 31.3 | **33.3** |
| | AMC23 | 70.5 | 70.5 | 71.0 | 69.5 | **75.5** |
| | GSM8K | 66.7 | 69.7 | 67.8 | **73.2** | 69.2 |
| | Math500 | 62.6 | 60.4 | 62.6 | **64.2** | 63.2 |
| Scientific QA | MMLU-Pro | 51.5 | 52.3 | 52.9 | 51.6 | **54.7** |
| | GPQA | 39.9 | 39.9 | 39.9 | 40.9 | **44.9** |
| Reasoning | MuSR | 52.6 | 51.9 | **54.2** | 52.5 | 54.1 |
| | DROP | 50.8 | **56.4** | 55.3 | 29.5 | 50.2 |
| | QASC | 82.1 | 81.6 | 83.2 | 80.8 | **84.1** |
| QA | 2wiki | 26.2 | 26.6 | 30.5 | 15.0 | **31.6** |
| | HotpotQA | 18.1 | 17.8 | 19.1 | 13.6 | **19.7** |
| QA-RAG | 2wiki_RAG | 36.7 | **39.3** | 39.1 | 20.2 | 37.9 |
| | HotpotQA_RAG | 27.1 | 27.1 | **27.9** | 17.9 | 27.3 |

Table 20: Ablation results of single dimension supervised training QWEN3-8B on RewardBench2. Training pairs are selected from **ANY** subset. **DRM** means training with DRM supervision. All training pairs are selected from **ANY** subset.

| Task Domain | Dataset | Native | Confidence | Coherence | Relevance | **DRM** |
|---|---|---|---|---|---|---|
| Code | CodeMMLU | 77.9 | 78.0 | 79.5 | 78.1 | **80.3** |
| | CodeScope | 86.5 | 86.8 | 86.3 | 86.7 | **87.4** |
| | Cruxeval | 91.6 | 92.9 | 91.8 | 91.9 | **93.0** |
| | Execution-v2 | 98.5 | 98.1 | 98.3 | 98.3 | **99.0** |
| Preference | RM-Bench | 85.4 | 84.8 | 84.6 | 84.8 | **85.6** |
| | UltraFeedback | 71.3 | 71.1 | 72.0 | 72.0 | **73.2** |
| Math | AIME24 | 38.0 | 38.7 | 39.3 | **45.3** | 44.7 |
| | AMC23 | 72.0 | 72.5 | 77.5 | **79.5** | 79.0 |
| | GSM8K | 95.6 | 95.2 | **95.7** | 95.4 | 96.1 |
| | Math500 | 73.2 | 72.4 | 74.6 | 74.2 | **75.6** |
| Scientific QA | MMLU-Pro | 65.3 | 62.5 | 70.5 | 70.7 | **71.4** |
| | GPQA | 48.0 | 43.9 | 54.0 | 56.1 | **58.1** |
| Reasoning | MuSR | 63.5 | 64.8 | 63.5 | 62.8 | **65.5** |
| | DROP | 74.7 | 74.4 | 74.0 | 74.6 | **74.9** |
| | QASC | 94.1 | 93.8 | 93.7 | 94.0 | **94.2** |
| QA | 2wiki | 39.8 | 40.9 | 39.5 | 42.0 | **42.2** |
| | HotpotQA | 29.2 | 28.3 | 28.7 | 27.3 | **29.4** |
| QA-RAG | 2wiki_RAG | 55.7 | 55.7 | 55.1 | 55.9 | **56.1** |
| | HotpotQA_RAG | 40.5 | 40.2 | 40.0 | 40.2 | **40.7** |

Table 21: Ablation results of single dimension supervised training LLAMA3.1-8B-INSTRUCT on HotpotQA with RAG. Training pairs are selected from **ANY** subset. **DRM** means training with DRM supervision. All training pairs are selected from **ANY** subset.

| Task Domain | Dataset | Native | Confidence | Coherence | Relevance | **DRM** |
|---|---|---|---|---|---|---|
| Code | CodeMMLU | 58.8 | 58.7 | 59.6 | 58.1 | **60.5** |
| | CodeScope | 34.8 | 37.6 | **41.8** | 39.3 | 41.7 |
| | Cruxeval | 50.4 | 54.5 | 54.0 | 52.5 | **56.2** |
| | Execution-v2 | 38.2 | 42.8 | **43.6** | 39.7 | 43.4 |
| Preference | RM-Bench | 56.4 | 59.9 | 60.3 | 59.4 | **62.9** |
| | UltraFeedback | 66.6 | 65.6 | 66.4 | 65.3 | **68.2** |
| Math | AIME24 | 4.7 | 3.3 | **6.7** | 5.3 | 3.3 |
| | AMC23 | 22.5 | 22.5 | 26.0 | 19.5 | **28.5** |
| | GSM8K | 88.8 | 87.8 | 89.6 | 90.6 | **91.5** |
| | Math500 | 39.6 | 41.2 | **46.2** | 41.8 | 45.0 |
| Scientific QA | MMLU-Pro | 41.9 | 46.2 | 47.9 | 46.5 | **49.6** |
| | GPQA | 31.3 | 28.3 | 29.3 | 31.8 | **34.3** |
| Reasoning | MuSR | 48.3 | 48.0 | 51.6 | 50.7 | **53.0** |
| | DROP | 56.9 | 62.3 | 65.3 | 58.9 | **67.3** |
| | QASC | 84.4 | 83.4 | 85.4 | 87.5 | 87.5 |
| QA | 2wiki | 33.8 | 35.2 | 38.4 | 37.6 | **40.9** |
| | HotpotQA | 29.3 | 31.7 | **33.2** | 30.7 | 32.8 |
| QA-RAG | 2wiki_RAG | 31.2 | 32.8 | 45.5 | 43.0 | **48.4** |
| | HotpotQA_RAG | 28.3 | 30.0 | 38.7 | 34.0 | **40.8** |

Table 22: Ablation results of single dimension supervised training R1-DISTIL-LLAMA8B on HotpotQA with RAG. Training pairs are selected from **ANY** subset. **DRM** means training with DRM supervision. All training pairs are selected from **ANY** subset.

| Task Domain | Dataset | Native | Confidence | Coherence | Relevance | DRM |
|---|---|---|---|---|---|---|
| Code | CodeMMLU | 59.7 | 64.2 | 64.6 | 65.4 | **66.6** |
| | CodeScope | 67.4 | 67.3 | 68.5 | 69.2 | **69.7** |
| | Cruxeval | 71.9 | 71.9 | 75.2 | 73.1 | **75.4** |
| | Execution-v2 | 80.8 | 80.8 | 83.5 | 81.8 | **85.6** |
| Preference | RM-Bench | 71.9 | 71.9 | 70.0 | 70.2 | **72.9** |
| | UltraFeedback | 65.2 | 65.0 | 64.0 | 64.4 | **67.0** |
| Math | AIME24 | 28.7 | 28.0 | **32.7** | 30.7 | 30.7 |
| | AMC23 | 70.5 | 64.0 | 79.5 | 77.5 | **80.5** |
| | GSM8K | 66.7 | 66.2 | 84.7 | **87.8** | 86.4 |
| | Math500 | 62.6 | 58.6 | 65.8 | 65.4 | **67.2** |
| Scientific QA | MMLU-Pro | 51.5 | 52.4 | 51.6 | 53.5 | **54.9** |
| | GPQA | 39.9 | **41.4** | 39.9 | 38.4 | **41.4** |
| Reasoning | MuSR | 52.6 | 54.8 | 56.0 | 57.5 | **58.6** |
| | DROP | 50.8 | 63.0 | 63.0 | 41.0 | **65.4** |
| | QASC | 82.1 | 84.6 | 84.1 | 83.1 | **85.2** |
| QA | 2wiki | 26.2 | 24.3 | 37.3 | 7.1 | **37.9** |
| | HotpotQA | 18.1 | 20.2 | 22.6 | 13.9 | **24.0** |
| QA-RAG | 2wiki_RAG | 36.7 | 46.0 | 49.6 | 25.6 | **51.6** |
| | HotpotQA_RAG | 27.1 | 33.5 | 35.8 | 22.6 | **37.2** |

Table 23: Ablation results of single dimension supervised training QWEN3-8Bon HotpotQA with RAG. Training pairs are selected from **ANY** subset. **DRM** means training with DRM supervision. All training pairs are selected from **ANY** subset.

| Task Domain | Dataset | Native | Confidence | Coherence | Relevance | DRM |
|---|---|---|---|---|---|---|
| Code | CodeMMLU | 77.9 | 77.3 | 79.0 | **79.2** | 79.0 |
| | CodeScope | 86.5 | 86.6 | 86.6 | 86.8 | **87.3** |
| | Cruxeval | 91.6 | 92.1 | 92.0 | 91.9 | **92.2** |
| | Execution-v2 | 98.5 | 98.1 | 97.9 | 98.3 | **98.7** |
| Preference | RM-Bench | 85.4 | **85.9** | 84.6 | 83.8 | 85.2 |
| | UltraFeedback | 71.3 | 72.0 | 71.4 | 72.2 | **72.7** |
| Math | AIME24 | 38.0 | 38.0 | 44.0 | 45.3 | **47.3** |
| | AMC23 | 72.0 | 73.0 | 78.5 | **83.5** | 82.5 |
| | GSM8K | 95.6 | 95.5 | 95.4 | 95.7 | **96.0** |
| | Math500 | 73.2 | 72.0 | 75.6 | 76.2 | **76.4** |
| Scientific QA | MMLU-Pro | 65.3 | 62.4 | **71.1** | 70.1 | 70.4 |
| | GPQA | 48.0 | 42.9 | 52.0 | 55.6 | **56.1** |
| Reasoning | MuSR | 63.5 | **63.9** | 62.7 | 63.2 | 63.5 |
| | DROP | **74.7** | 73.2 | 73.5 | 74.4 | 74.2 |
| | QASC | **94.1** | 92.7 | 93.7 | 94.0 | 94.0 |
| QA | 2wiki | 39.8 | **40.9** | 38.9 | 37.5 | 40.1 |
| | HotpotQA | **29.2** | 27.2 | 28.5 | 28.2 | 28.7 |
| QA-RAG | 2wiki_RAG | 55.7 | 55.5 | 55.5 | 53.9 | **56.9** |
| | HotpotQA_RAG | **40.5** | 38.7 | 38.1 | 35.4 | 38.8 |

