# OpenReview forum: "From Answer to Think: Multidimensional Supervision of Reasoning Process for LLM Optimization"
_ICLR.cc/2026/Conference — Submitted to ICLR 2026_

### Official Review · Reviewer_pnbE · 2025-10-26

**Soundness:** 3
**Presentation:** 3
**Contribution:** 2
**Rating:** 4
**Confidence:** 4

**Summary:**

This paper introduces DRM , a new framework that supervises large language models’ reasoning processes across three interpretable dimensions—confidence, relevance, and coherence—to more effectively improve reasoning ability and generalization than traditional outcome-only training methods.

**Strengths:**

- The idea is simple and reasonable. Experiments show the effectiveness of DRM.

- DRM can simple replace what orm did in different algorithm.

**Weaknesses:**

- In table 3, the improvement is limited, not sure whether the improvement only exist in given hyperparameter.

**Questions:**

- In the paper, the weights for the three dimensions are tuned empirically or via grid search. Were these weights optimized separately for each task, or were they fixed across all tasks? Also, did you compare whether different tasks (e.g., math vs. code generation) require different optimal weight configurations?

- The multi-dimensional supervision seems to rely on several external evaluators. Could you clarify how much additional computational overhead or latency this introduces during training and inference?

- Did you conduct any analysis on how DRM affects the reasoning process qualitatively — for instance, does it make the model’s explanations longer, more structured, or more confident?

---

> ### Author Response · Authors · 2025-11-21
> **Rebuttal to Reviewer pnbE (Part 1/4)**
>
> (Due to the character limit, we split our response into four parts. This is Part 1.)
>
> ---
>
> We sincerely thank you for reviewing our work and for providing thoughtful comments, which helped us improve both the clarity and completeness of the paper. We appreciate the time taken to review our manuscript and have addressed the raised points as follows.
>
> ## Question.1
>
> > In the paper, the weights for the three dimensions are tuned empirically or via grid search. Were these weights optimized separately for each task, or were they fixed across all tasks? Also, did you compare whether different tasks (e.g., math vs. code generation) require different optimal weight configurations?
>
> We apply a fixed set of weights across all tasks and datasets in the main experiments and do not optimize them individually for specific tasks. This decision rests on the empirical robustness of our framework, as detailed below.
>
> **Our method is robust to model backbones**. In Section3.2, we conduct a comprehensive ablation study on different aggregation methods. We examine the performance of each dimension (Confidence, Relevance, Coherence) independently and compare them against combined methods, utilizing both equal weighting and our proposed fixed weighting schemes. We select the final model output based on the highest process reward under each setting and measure answer correctness. The results in Table3 demonstrate that combining dimensions consistently outperforms single-dimension metrics, and the performance remains robust across different weighting schemes and model backbones. The marginal difference between fixed weights and alternative configurations suggests that the method is insensitive to precise weight tuning.
>
> > Table3 (Page5)
> >
> > Answer correctness (\%) of DRM construction approaches on RewardBench2. Native means the performance of the backbone models. (0.1,0.2,0.7) means weights for Confidence, Relevance and Coherence are 0.1, 0.2, 0.7, respectively. LTR denotes that a Learning-to-Rank model is employed for the construction. The highest result in each row is in **bold**.
> > | Model | Native | Confidence | Relevance | Coherence | Weighted Equally | Weighted (0.1,0.2,0.7) | LTR |
> > | :------------------- | :----- | :--------- | :-------- | :-------- | :--------------- | :--------------------- | :-------- |
> > | LLaMA3.1-8B-Instruct | 67.17 | 65.44 | 72.32 | 78.55 | 77.45 | 78.57 | **79.13** |
> > | R1-Distil-Llama8B | 63.46 | 63.10 | 66.76 | **76.35** | 75.11 | 76.16 | 75.18 |
> > | Qwen3-8B | 84.87 | 83.20 | 85.10 | 85.54 | 85.01 | 85.65 | **85.88** |
>
> **Our method is robust across tasks and training data distributions**. In Appendix G.1, we further verify this robustness on the HotpotQA with RAG dataset to assess generalization across different data distributions. As shown in Table11, the combined methods maintain consistent effectiveness compared to individual dimensions, reinforcing the stability and robustness of our design. The fixed weighting scheme performs comparably to other configurations, which confirms that distinct tasks do not require specialized weight tuning to achieve optimal performance.
>
> > Table11 (Page27)
> >
> > Answer correctness (\%) of DRM construction approaches on HotpotQA with RAG. Native means the performance of the backbone models. (0.1,0.2,0.7) means weights for Confidence, Relevance and Coherence are 0.1, 0.2, 0.7, respectively. LTR denotes that a Learning-to-Rank model is employed for the construction. The highest result in each row is in **bold**.
> > | Model | Native | Confidence | Relevance | Coherence | Weighted Equally | Weighted (0.1,0.2,0.7) | LTR |
> > | :--- | :--- | :--- | :--- | :--- | :--- | :--- | :--- |
> > | LLaMA3.1-8B-Instruct | 45.31 | 52.42 | 54.56 | 61.36 | 61.33 | **61.70** | 61.25 |
> > | R1-Distil-Llama8B | 43.09 | 49.77 | 47.90 | 55.58 | 55.49 | 55.58 | **55.76** |
> > | Qwen3-8B | 63.61 | 63.37 | 64.36 | 64.31 | **64.55** | 64.39 | 64.11 |
>
> **Our method is robust to weight calibration methods**. To further address your concern on our integration methods, we also conduct an additional experiment to explore moving beyond linear weight calibration to learnable weighting mechanisms. In particular, we implement a Learning-to-Rank (LTR) approach, treating the dimension-wise scores as features and framing selection as a reranking task. LTR selects the reasoning process which ranks at the top. The results of this experiment are listed in the "LTR" column of Table3 and Table11. We observe that despite employing different weighting strategies, the results remain consistently similar across backbone models and training datasets. This consistency further indicates the **robustness of our overall framework**.
>
> For clarity and simplicity, we therefore retain the linear-weight combined methods in the main paper, as we believe a stable and interpretable weighting approach is more "elegant", particularly when learnable methods (LTR) do not yield noticeable gains.

---

> ### Author Response · Authors · 2025-11-21
> **Rebuttal to Reviewer pnbE (Part 2/4)**
>
> (Due to the character limit, we split our response into four parts. This is Part 2.)
>
> ---
>
> ### Weakness.1
>
> > In Table3, the improvement is limited, not sure whether the improvement only exist in given hyperparameter.
>
> **Table3 does not represent the final model performance.** Instead, it evaluates the accuracy of answers derived from reasoning paths selected during training data construction. **While the margins in this intermediate step appear limited, the final model trained on this data yields substantial and consistent improvements across various downstream tasks (shown in Table5)**, which confirms the **effectiveness of our approach**.
>
> > Table5 (Page8)
> >
> > Results of off-policy DPO with SFT loss training. RLPR and Klear are baseline models that share the same backbone architectures as their respective counterparts. Skywork indicates that the model's training set is constructed using Skywork reward model. DRM represents models trained by DRM supervision. For each row within a model group, the highest score is in **bold**.
> > | - | **Backbone** | LLaMA | LLaMA | LLaMA | LLaMA | DSLLaMA| DSLLaMA | DSLLaMA | Qwen | Qwen | Qwen | Qwen |
> > | :---------------- | :------------ | :----: | :-----: | :-----: | :------: | :----: | :------: | :------: | :----: | :------: | :-----: | :------: |
> > | **Task Domain** | **Dataset** | Native | RLPR | Skywork | DRM | Native | Skywork | DRM | Native | Klear | Skywork | DRM |
> > | **Code** | CodeMMLU | 58.8 | 58.0 | 57.6 | **59.9** | 59.7 | 62.9 | **66.3** | 77.9 | 77.4 | 79.3 | **80.3** |
> > | **Code** | CodeScope | 34.8 | 38.7 | 39.3 | **41.1** | 67.4 | 68.2 | **70.2** | 86.5 | **88.1** | 86.2 | 87.4 |
> > | **Code** | Cruxeval | 50.4 | 53.6 | 53.6 | **57.5** | 71.9 | 77.0 | **77.2** | 91.6 | 87.2 | 91.9 | **93.0** |
> > | **Code** | Execution-v2 | 38.2 | 44.7 | 42.8 | **45.3** | 80.8 | 82.0 | **86.0** | 98.5 | 95.2 | 97.9 | **99.0** |
> > | **Preference** | RM-Bench | 56.4 | 60.2 | 59.8 | **61.0** | 71.9 | 73.4 | **74.6** | 85.4 | 83.7 | 85.1 | **85.6** |
> > | **Preference** | UltraFeedback | 66.6 | 68.5 | 67.0 | **69.9** | 65.2 | 66.5 | **66.8** | 71.3 | 68.1 | 72.2 | **73.2** |
> > | **Math** | AIME24 | 4.7 | **6.0** | 4.0 | **6.0** | 28.7 | 26.7 | **33.3** | 38.0 | 40.0 | 38.7 | **44.7** |
> > | **Math** | AMC23 | 22.5 | 26.0 | 25.5 | **29.5** | 70.5 | 74.5 | **75.5** | 72.0 | 75.0 | 76.0 | **79.0** |
> > | **Math** | GSM8K | 88.8 | 90.0 | 89.8 | **91.8** | 66.7 | **73.7** | 69.2 | 95.6 | 93.8 | 95.8 | **96.1** |
> > | **Math** | Math500 | 39.6 | 47.2 | 42.6 | **48.4** | 62.6 | **65.6** | 63.2 | 73.2 | 68.2 | 72.6 | **75.6** |
> > | **Scientific QA** | MMLU-Pro | 41.9 | 36.3 | 46.7 | **48.7** | 51.5 | 52.8 | **54.7** | 65.3 | 67.1 | 70.0 | **71.4** |
> > | **Scientific QA** | GPQA | 31.3 | 30.8 | 33.3 | **35.9** | 39.9 | 37.4 | **44.9** | 48.0 | 55.6 | 52.5 | **58.1** |
> > | **Reasoning** | MuSR | 48.3 | 48.7 | 49.7 | **51.7** | 52.6 | 52.8 | **54.1** | 63.5 | 50.8 | 63.5 | **65.5** |
> > | **Reasoning** | DROP | 56.9 | 45.4 | 63.0 | **63.6** | 50.8 | **54.5** | 50.2 | 74.7 | 68.8 | 74.2 | **74.9** |
> > | **Reasoning** | QASC | 84.4 | 87.0 | 87.1 | **87.2** | 82.1 | 82.5 | **84.1** | 94.1 | 93.3 | 93.7 | **94.2** |
> > | **QA** | 2wiki | 33.8 | 32.1 | 32.4 | **35.6** | 26.2 | 29.3 | **31.6** | 39.8 | 35.9 | 40.0 | **42.2** |
> > | **QA** | HotpotQA | 29.3 | 29.9 | 30.4 | **31.8** | 18.1 | 19.3 | **19.7** | 29.2 | 19.6 | 29.1 | **29.4** |
> > | **QA-RAG** | 2wiki_RAG | 31.2 | 38.7 | 34.8 | **39.9** | 36.7 | **39.2** | 37.9 | 55.7 | 52.2 | 55.8 | **56.1** |
> > | **QA-RAG** | HotpotQA_RAG | 28.3 | 32.8 | 33.2 | **34.5** | 27.1 | 26.5 | **27.3** | 40.5 | 34.3 | 40.3 | **40.7** |

---

> ### Author Response · Authors · 2025-11-21
> **Rebuttal to Reviewer pnbE (Part 3/4)**
>
> (Due to the character limit, we split our response into four parts. This is Part 3.)
>
> ---
>
> ## Question.2
>
> > The multi-dimensional supervision seems to rely on several external evaluators. Could you clarify how much additional computational overhead or latency this introduces during training and inference?
>
> We thank the reviewer for this question. To clarify potential confusion: the multi-dimensional supervision is strictly a training-time mechanism. Consequently, the trained model operates independently of external evaluators, **incurring zero extra latency or computational overhead during inference**.
>
> Regarding training, we conduct an additional computation analysis in Appendix E. We find that the full DRM implementation requires approximately 160% more GPU-hours than RLVR. Therefore, we introduce **DRM-Light** as an efficient alternative for budget-constrained scenarios. By replacing the coherence evaluator with a lightweight ORM (Skywork-Reward-V2-Llama-3.1-8B, also mentioned in Table5), **DRM-Light** requires only **25% more GPU** allocation and **extends training time by just 9%**, resulting in a total cost of 136% GPU hours relative to RLVR. The performance results are shown in Table9. As summarized in Table10, this variant maintains strong performance, offering a highly cost-effective **balance** between computational efficiency and model performance.
>
> > Table10 (Page23)
> >
> > Comparison of computational overhead and performance trade-offs on R1-Distil-Llama8B. The performance reported is the average on Code, Math, Scientific QA and Reasoning benchmarks.
> > | Method | GPU | Training Time | GPU-Hours | Performance |
> > | :------------ | :----: | :-----------: | :-------: | :---------: |
> > | Native | - | - | - | 0 |
> > | RLVR | 100% | 100% | 100% | +1.12 |
> > | **DRM** | 162.5% | 160% | 260% | +4.19 |
> > | **DRM-Light** | 125% | 109% | 136% | +2.33 |
>
> > Table9 (Page23)
> >
> > Results of R1-Distil-Llama8B on-policy GRPO training evaluated on Code, Math, Scientific QA and Reasoning benchmarks. For each row, the highest score is in **bold**.
> > | Task Domain | Dataset | Native | RLVR | DRM | DRM-light |
> > | :---------------- | :----------- | :------: | :------: | :------: | :-------: |
> > | **Code** | CodeMMLU | 62.1 | 62.4 | **65.1** | 63.9 |
> > | **Code** | CodeScope | 66.4 | **69.2** | 68.2 | 67.9 |
> > | **Code** | CruxEval | 74.3 | 74.4 | **76.0** | 73.4 |
> > | **Code** | Execution-v2 | 82.9 | 82.3 | 83.5 | **86.4** |
> > | **Math** | AIME24 | 27.3 | 29.3 | **34.7** | 26.7 |
> > | **Math** | AMC23 | 70.0 | 70.5 | **77.5** | 73.0 |
> > | **Math** | GSM8K | 66.8 | 72.5 | **83.1** | 81.2 |
> > | **Math** | MATH500 | 61.2 | 62.8 | **67.0** | 65.8 |
> > | **Scientific QA** | MMLU-Pro | **53.7** | 53.6 | 53.4 | 52.9 |
> > | **Scientific QA** | GPQA | 39.9 | 39.4 | **43.9** | 40.4 |
> > | **Reasoning** | MuSR | 52.3 | **53.0** | **53.0** | 51.2 |
> > | **Reasoning** | QASC | 82.9 | 83.8 | 84.6 | **84.9** |

---

> ### Author Response · Authors · 2025-11-21
> **Rebuttal to Reviewer pnbE (Part 4/4)**
>
> (Due to the character limit, we split our response into four parts. This is Part 4.)
>
> ---
>
> ## Question.3
>
> > Did you conduct any analysis on how DRM affects the reasoning process qualitatively — for instance, does it make the model’s explanations longer, more structured, or more confident?
>
> Yes, we have conducted such an analysis.
>
> In Section4.1, we analyze the qualitative impact of DRM and observe that it fosters a rigorous and systematic reasoning style beyond mere accuracy improvements. Regarding **logical soundness**, our evaluation using GPT-4o reveals that DRM significantly reduces instances of "correct answers derived from flawed reasoning" compared to RLVR (see Figure3a (Page9), here we summarize as Table of data shown in Figure3a (Page9)). We also conduct a **case study** in Appendix C, where a sample from RLVR and a sample from DRM are compared. This comparison shows that cases sampled by RLVR can produce correct answers but with flawed reasoning, while cases selected by DRM provide both correct answers and sound reasoning. **This case study indicates that our supervision approach can reduce the occurrence of correct answer with flawed reasoning cases**.
>
> Furthermore, to address your concern on the quality impact, we conduct additional experiments. The results of pairwise judgments by GPT-4o indicate that DRM induces a noticeably **more structured** reasoning pattern compared to backbone models across various benchmarks (see Figure3b (Page9), here we summarize as Table of data shown in Figure3b (Page9)). This structural enhancement accompanies **a modest increase in reasoning length (+4.9%)**, which remains more efficient than the increase observed in RLVR (+12.1%). The results reflect a positive behavioral shift: the model engages in thorough, step-by-step derivation to ensure precision, effectively avoiding reasoning shortcuts while maintaining necessary conciseness.
>
> > Table of data shown in Figure3a (Page9).
> >
> > Count of correct answers with flawed reasoning as evaluated by GPT-4o. Each training set contains approximately 6000 samples.
> > | Model | RLVR | DRM |
> > | :--- | :--- | :--- |
> > | **LLaMA3.1-8B-Instruct** | 734 | 298 |
> > | **R1-Distil-Llama8B** | 227 | 152 |
> > | **Qwen3-8B** | 972 | 427 |
>
> > Table of data shown in Figure3b (Page9).
> >
> > Comparison of reasoning process structure between R1-Distil-Llama8B and its DRM-supervised variants as evaluated by GPT-4o.
> > | Benchmark | Native win rate | DRM win rate |
> > | :--- | :--- | :--- |
> > | **RewardBench 2** | 35% | 65% |
> > | **CodeScope** | 35% | 65% |
> > | **Execution-v2** | 32% | 68% |
> > | **AMC23** | 29% | 71% |
> > | **Math500** | 32% | 68% |
> > | **MMLU-Pro** | 35% | 65% |
> > | **GPQA** | 29% | 71% |
> > | **MuSR** | 37% | 63% |
>
> > Average output length of R1-Distil-Llama8B, tested on the sum of all benchmarks.
> > |-| Native | RLVR | DRM |
> > | :--- | :--- | :--- | :--- |
> > | Average output length | 3874.3 | 4341.8 | 4062.7 |
> > | Ratio | - | +12.1% | +4.9% |
>
> ---
>
> We hope that our responses and the additional evidence presented here make the contributions and technical soundness of the work clearer. We have uploaded a revised version of our paper, in which the newly added content is highlighted in blue for convenience. If you have any further questions or would like to discuss anything in more detail, we’d be very happy to continue the conversation.

---

> ### Author Response · Authors · 2025-11-27
> **Following up on our response to your concerns**
>
> Dear Reviewer pnbE,
>
> Thank you for your time and constructive comments.
>
> As the discussion period is ending soon (Dec 3rd), we would like to kindly check if our previous response and the revised paper have addressed your concerns.
>
> In our rebuttal, we have made specific efforts to clarify the points you raised (especially regarding the integration of dimensions, the clarification of computational overhead, and the in‑depth analysis on how DRM affects the reasoning process qualitatively). We hope the new results and explanations demonstrate the soundness and contribution of our work.
>
> We value your feedback and would appreciate it if you could let us know if there are any concerns. We are happy to engage in further discussion.
>
> Best regards,
>
> The Authors

---

> > ### Comment · Reviewer_pnbE · 2025-11-27
> > **The timing of your follow-up… is quite tricky.**
> >
> > First, regarding your response: I really appreciate the additional experiments you added to further demonstrate the effectiveness of your work. However, it seems that the extra time required for DRM is a major issue, and the performance of your proposed DRM-light improves very little compared to RLVR. Most of the average gains you report still come almost entirely from that single gsm8k. The remaining two points you clarified did resolve my concerns.
> >
> > Regarding the light version, I have a question. You mentioned that you launched an additional 8B reward model, but why does the GPU usage increase by only about 25%? In your experiments, were you measuring only the GPU usage of the DRM training itself, or did you also include the extra GPU cost introduced by running the additional reward model?
> >
> > As for changing my score, I’m not planning to revise it before ICLR announces its official resolution to this incident. Sorry about that. But AC could see my score as 5 at least.

---

> > > ### Author Response · Authors · 2025-11-28
> > >
> > > Thank you very much for your constructive comments and for raising this important point. We truly appreciate the opportunity to clarify the details of our experimental setup.
> > >
> > > Regarding your concern on resource usage: **We confirm that the reported resource usage includes the entire pipeline, covering both the main training process and the inference costs of the additional reward models.** The GPU resource increase of only 25% is due to the scale of our baseline experiment. Here is the detailed breakdown:
> > >
> > > 1.  Baseline (RLVR): We use **64 GPUs** (8 nodes $\times$ 8 GPUs) for the main training process.
> > > 2.  DRM: We add 4 extra nodes for the Coherence evaluator and 1 extra node for the Relevance evaluator.
> > >     - Total GPUs: $64 + 32 + 8 = 104$ GPUs.
> > >     - Ratio: $104 / 64 = 1.625 $.
> > > 3.  DRM-light: We replace the Coherence evaluator with a smaller one, enabling it to run on just **1 extra node (8 GPUs)** instead of 4. We continue to use 1 extra node for Relevance.
> > >     - Total GPUs: $64 + 8 + 8 = 80$ GPUs.
> > >     - Ratio: $80 / 64 = 1.25$. **This results in exactly a 25% increase.**
> > >
> > > We understand why this might seem surprising. If we were training on a single machine, adding an 8B model would indeed be a heavy burden. However, since our setup already requires a substantial cluster, adding 16 GPUs for the evaluators is a relatively small percentage increase. Additionally, using fewer machines in DRM-light reduces communication overhead, which also helps with the total training time.
> > >
> > > This multi-machine configuration is documented in the `readme.md` file (Line 87) of our **original supplementary materials** (submitted with the initial paper). Here we quote:
> > >
> > > > - recommended: 4-multiples machines x 8 GPUs each to run this script
> > >
> > > We hope this explanation clarifies the resource calculation. Please let us know if there are any other details we can provide.
> > >
> > > Best regards,
> > >
> > > Authors

---

### Official Review · Reviewer_ubnr · 2025-10-30

**Soundness:** 2
**Presentation:** 3
**Contribution:** 2
**Rating:** 4
**Confidence:** 3

**Summary:**

This paper proposes a Dimension-level Reward Model (DRM) that scores a model’s reasoning process along three complementary dimensions—Confidence, Relevance, and Coherence—and uses this multidimensional signal to supervise both off-policy (DPO + SFT) and on-policy (GRPO-style) optimization. Unlike answer-only reward schemes (RLVR) that deliver sparse, outcome-level feedback and often reward “correct answer, flawed reasoning,” and unlike process-level reward models (PRMs) that require task-specific step segmentation, DRM delivers dense, interpretable, ground-truth-free rewards over the entire chain of thought.

**Strengths:**

1. DRM directly targets two known gaps—sparse/answer-only rewards and PRM step-segmentation requirements—by shifting to dimension-level scoring that is dense, ground-truth-free, and interpretable.

2. The DRM reward can be integrated with standard training. It supervises off-policy DPO+SFT (pair selection) and augments on-policy GRPO (added advantage).

3. Across diverse tasks and backbones, DRM-supervised models outperform native and strong baselines.

**Weaknesses:**

1. Relevance depends on a reranker and Coherence on an ORM; the paper fixes dimension weights via grid search. While practical, robustness to judge/model choice and weight calibration is under-analyzed.

2. Using log-prob as self-confidence is intuitive, but there’s limited study of calibration across domains/backbones or under distribution shift, and little comparison to alternative confidence estimators.

3. In GRPO combinations, a few reasoning-heavy or knowledge-intensive datasets see slight regressions vs. DRM alone (e.g., MuSR/GPQA), suggesting interaction effects between answer-only and reasoning rewards that merit deeper analysis.

**Questions:**

Please see the weakness.

---

> ### Author Response · Authors · 2025-11-21
> **Rebuttal to Reviewer ubnr (Part 1/3)**
>
> (Due to the character limit, we split our response into three parts. This is Part 1.)
>
> ---
>
> We sincerely thank you for reviewing our work and for providing thoughtful comments, which helped us improve both the clarity and completeness of the paper. We appreciate the time taken to review our manuscript and have addressed the raised points as follows.
>
> ## Question.1
>
> > Relevance depends on a reranker and Coherence on an ORM; the paper fixes dimension weights via grid search. While practical, robustness to judge/model choice and weight calibration is under-analyzed.
>
> In Section3.2, we conduct a comprehensive ablation study on different aggregation methods to verify whether our method is robust to model choice. We examine the performance of each dimension (Confidence, Relevance, Coherence) independently and compare them against combined methods, utilizing both equal weighting and our proposed fixed weighting schemes. We select the final model output based on the highest process reward under each setting and measure answer correctness. The results in Table3 demonstrate that combining dimensions consistently outperforms single-dimension metrics, and the **performance remains robust across different weighting schemes and model backbones**.
>
> > Table3 (Page5)
> >
> > Answer correctness (\%) of DRM construction approaches on RewardBench2. Native means the performance of the backbone models. (0.1,0.2,0.7) means weights for Confidence, Relevance and Coherence are 0.1, 0.2, 0.7, respectively. LTR denotes that a Learning-to-Rank model is employed for the construction. The highest result in each row is in **bold**.
> > | Model | Native | Confidence | Relevance | Coherence | Weighted Equally | Weighted (0.1,0.2,0.7) | LTR |
> > | :------------------- | :----- | :--------- | :-------- | :-------- | :--------------- | :--------------------- | :-------- |
> > | LLaMA3.1-8B-Instruct | 67.17 | 65.44 | 72.32 | 78.55 | 77.45 | 78.57 | **79.13** |
> > | R1-Distil-Llama8B | 63.46 | 63.10 | 66.76 | **76.35** | 75.11 | 76.16 | 75.18 |
> > | Qwen3-8B | 84.87 | 83.20 | 85.10 | 85.54 | 85.01 | 85.65 | **85.88** |
>
> In Appendix G.1, we further verify this robustness on the HotpotQA with RAG dataset to assess **robustness across different data distributions**. As shown in Table11, the combined methods maintain consistent effectiveness compared to individual dimensions, reinforcing the stability and robustness of our design.
>
> > Table11 (Page27)
> >
> > Answer correctness (\%) of DRM construction approaches on HotpotQA with RAG. Native means the performance of the backbone models. (0.1,0.2,0.7) means weights for Confidence, Relevance and Coherence are 0.1, 0.2, 0.7, respectively. LTR denotes that a Learning-to-Rank model is employed for the construction. The highest result in each row is in **bold**.
> > | Model | Native | Confidence | Relevance | Coherence | Weighted Equally | Weighted (0.1,0.2,0.7) | LTR |
> > | :--- | :--- | :--- | :--- | :--- | :--- | :--- | :--- |
> > | LLaMA3.1-8B-Instruct | 45.31 | 52.42 | 54.56 | 61.36 | 61.33 | **61.70** | 61.25 |
> > | R1-Distil-Llama8B | 43.09 | 49.77 | 47.90 | 55.58 | 55.49 | 55.58 | **55.76** |
> > | Qwen3-8B | 63.61 | 63.37 | 64.36 | 64.31 | **64.55** | 64.39 | 64.11 |
>
> To address your concern on weight calibration methods, we also conduct an additional experiment to explore moving **beyond linear weight calibration to learnable weighting mechanisms**. In particular, we implement a Learning-to-Rank (LTR) approach, treating the dimension-wise scores as features and framing selection as a reranking task. LTR selects the reasoning process which ranks at the top. The results of this experiment are listed in the "LTR" column of Table3 and Table11. Together with the linear-weight experiments, we observe that **despite employing different weighting strategies, the results remain consistently similar across backbone models and training datasets**. This consistency further indicates the **robustness of our overall framework**.
>
> For clarity and simplicity, we therefore retain the linear-weight combined methods in the main paper, as we believe a stable and interpretable weighting approach is more "elegant", particularly when learnable methods (LTR) do not yield noticeable gains.

---

> ### Author Response · Authors · 2025-11-21
> **Rebuttal to Reviewer ubnr (Part 2/3)**
>
> (Due to the character limit, we split our response into three parts. This is Part 2.)
>
> ---
>
> ## Question.2
>
> > Using log-prob as self-confidence is intuitive, but there’s limited study of calibration across domains/backbones or under distribution shift, and little comparison to alternative confidence estimators.
>
> In Section3.2, we conduct comprehensive experiments on three distinct backbones: LLaMA3.1-8B-Instruct, R1-Distil-Llama8B, and Qwen3-8B to verify whether our Confidence score is robust across different backbone models. We select the final model output based on the Confidence score and measure answer correctness. The results are shown in Table3 (Page5). The results show that our **Confidence score is robust across different backbone models**.
>
> In Appendix G.1, we conduct our experiment on the HotpotQA with RAG dataset to verify whether our Confidence score is robust across different training data distributions/domains. The results are shown in Table11 (Page27). The results show that our **Confidence score is robust across varying training data distributions/domains**.
>
> To address your concern on alternative confidence estimators, we conduct additional experiments in Appendix D on two alternative Confidence estimators: perplexity and average token entropy. The former uses only the next token log-probability, similar to our method, while the latter aggregates log-probabilities over all vocabulary tokens. They are defined as follows.
>
> $$
> \text{PPL}(X) = \exp\left( -\frac{1}{N} \sum_{i=1}^{N} \log P(x_i \mid x_{<i}) \right),
> $$
>
> where $P(x_i \mid x_{<i})$ denotes the probability of the $i$-th token $x_i$ given the preceding context $x_{<i}$. Intuitively, a lower perplexity indicates that the model assigns higher probabilities to the generated tokens, corresponding to higher confidence.
>
> $$
> \text{Entropy}(X) = \frac{1}{N} \sum_{i=1}^{N} H(P(\cdot \mid x_{<i})) = \frac{1}{N} \sum_{i=1}^{N} \left( - \sum_{v \in \mathcal{V}} P(v \mid x_{<i}) \log P(v \mid x_{<i}) \right),
> $$
>
> where $\mathcal{V}$ represents the model's vocabulary. High entropy implies a flat distribution where the model is uncertain among multiple choices, whereas low entropy indicates a peaked distribution where the model is confident in its prediction.
>
> We conduct comprehensive experiments on the two alternative confidence estimators. Specifically, we examine their correlation with answer correctness (results in Table7) and evaluate their performance under supervised training (results in Table8). As shown in the results, neither alternative outperforms our proposed approach. Therefore, these empirical results validate our original design choice, and we retain the current Confidence scoring approach in the main paper.
>
> > Table7 (Page21)
> >
> > Answer correctness (\%) of Confidence implementations on RewardBench2
> > Native means the performance of the backbone models. The highest result in each row is in **bold**.
> > | Model | Native | Confidence | Confidence$^{PPL}$ | Confidence$^{Entropy}$ |
> > | :--- | :--- | :--- | :--- | :--- |
> > | LLaMA3.1-8B-Instruct | 67.17 | 65.44 | **70.16** | 66.45 |
> > | R1-Distil-Llama8B | **63.46** | 63.10 | 62.28 | 61.30 |
> > | Qwen3-8B | **84.87** | 83.20 | 83.95 | 83.79 |

---

> ### Author Response · Authors · 2025-11-21
> **Rebuttal to Reviewer ubnr (Part 3/3)**
>
> (Due to the character limit, we split our response into three parts. This is Part 3.)
>
> ---
>
> > Table8 (Page22)
> >
> > Results of off-policy DPO with SFT loss training. For each row within a model group, the highest score is in **bold**.
> > | Task Domain | Dataset | LLaMA Native | LLaMA Conf | LLaMA PPL | LLaMA Entropy | DSLLaMA Native | DSLLaMA Conf | DSLLaMA PPL | DSLLaMA Entropy | Qwen Native | Qwen Conf | Qwen PPL | Qwen Entropy |
> > | :--- | :--- | :---: | :---: | :---: | :---: | :---: | :---: | :---: | :---: | :---: | :---: | :---: | :---: |
> > | **Code** | CodeMMLU | 58.8 | **59.1** | 54.9 | 54.6 | 59.7 | 62.3 | **62.4** | 61.0 | 77.9 | 78.8 | 76.8 | **78.9** |
> > | **Code** | CodeScope | 34.8 | **41.1** | 36.5 | 35.6 | **67.4** | 66.7 | 67.0 | 61.2 | 86.5 | 87.2 | 85.4 | **87.9** |
> > | **Code** | Cruxeval | 50.4 | **55.0** | 45.2 | 46.5 | 71.9 | **74.1** | 73.5 | 72.5 | 91.6 | **93.6** | 92.0 | 91.6 |
> > | **Code** | Execution-v2 | 38.2 | **41.8** | 40.1 | 38.6 | 80.8 | **82.9** | 82.3 | 79.1 | 98.5 | **98.7** | 98.3 | 97.7 |
> > | **Preference** | RM-Bench | 56.4 | **59.4** | 50.6 | 54.0 | 71.9 | 72.0 | **74.6** | 70.9 | **85.4** | 85.2 | 79.6 | 83.8 |
> > | **Preference** | UltraFeedback | 66.6 | **66.8** | 58.2 | 60.9 | 65.2 | **66.3** | 65.4 | 62.5 | 71.3 | **72.1** | 63.5 | 71.7 |
> > | **Math** | AIME24 | **4.7** | **4.7** | 4.0 | 2.7 | **28.7** | 27.3 | 27.3 | 26.0 | 38.0 | 40.7 | **42.7** | 42.0 |
> > | **Math** | AMC23 | 22.5 | **23.0** | 19.0 | 22.0 | 70.5 | **72.5** | 65.5 | 69.0 | 72.0 | 73.5 | 71.0 | **78.5** |
> > | **Math** | GSM8K | **88.8** | 83.0 | 71.3 | 68.7 | 66.7 | **69.7** | 67.9 | 67.3 | 95.6 | **96.2** | 95.2 | 95.1 |
> > | **Math** | Math500 | 39.6 | **41.8** | 34.8 | 34.2 | **62.6** | 62.2 | 61.0 | 60.6 | 73.2 | **73.8** | 72.4 | 73.0 |
> > | **Scientific QA** | MMLU-Pro | 41.9 | **47.1** | 35.8 | 39.3 | 51.5 | **52.5** | 51.9 | 51.8 | **65.3** | 62.8 | 60.1 | 64.8 |
> > | **Scientific QA** | GPQA | 31.3 | **32.8** | 28.8 | 26.8 | 39.9 | **42.9** | 42.4 | 35.4 | 48.0 | **48.5** | 41.4 | 47.5 |
> > | **Reasoning** | MuSR | 48.3 | **50.7** | 42.2 | 46.8 | 52.6 | 53.3 | **53.4** | 51.6 | 63.5 | **65.1** | 64.2 | 62.8 |
> > | **Reasoning** | DROP | **56.9** | 52.9 | 32.6 | 26.5 | 50.8 | **59.2** | 56.5 | 55.1 | 74.7 | **74.9** | 74.0 | 74.4 |
> > | **Reasoning** | QASC | **84.4** | 84.3 | 71.4 | 74.3 | 82.1 | **84.4** | 81.3 | 79.5 | **94.1** | **94.1** | 93.4 | 93.4 |
> > | **QA** | 2wiki | 33.8 | **35.8** | 20.6 | 18.1 | 26.2 | **28.1** | **28.1** | 24.1 | 39.8 | **42.3** | 38.3 | 40.9 |
> > | **QA** | HotpotQA | 29.3 | **30.0** | 21.8 | 21.0 | 18.1 | 18.7 | **19.3** | 17.3 | **29.2** | 29.1 | 26.7 | 28.1 |
> > | **QA-RAG** | 2wiki_RAG | **31.2** | 28.7 | 14.2 | 13.6 | 36.7 | **41.1** | 39.8 | 37.9 | 55.7 | 55.9 | **56.1** | 55.8 |
> > | **QA-RAG** | HotpotQA_RAG | **28.3** | **28.3** | 16.9 | 16.6 | 27.1 | 28.7 | **29.4** | 27.6 | 40.5 | 40.3 | **40.6** | 39.4 |
>
> ---
>
> ## Question.3
>
> > In GRPO combinations, a few reasoning-heavy or knowledge-intensive datasets see slight regressions vs. DRM alone (e.g., MuSR/GPQA), suggesting interaction effects between answer-only and reasoning rewards that merit deeper analysis.
>
> We hypothesize that the slight drop in Combination-supervised training performance stems from the distinct optimization objectives of **DRM and RLVR occasionally diverging**. To verify this, additionally, we visualize the training dynamics of R1-Distil-Llama8B in Figure2 (Page7) by plotting DRM versus RLVR rewards for each training batch in Section3.4. While the data shows a general positive correlation, distinct **outliers** reveal instances of signal misalignment. These outliers indicate conflicting supervision signals, **suggesting that outcome-level and process-level objectives pull the model in competing directions during complex reasoning**. This interference explains why the Combination method underperforms relative to the consistent process-level supervision of DRM. We include this analysis in the revised paper (see Page8.Line408). Here we quote:
>
> > _We provide empirical evidence for this interference in Figure2, illustrating the correlation between DRM and RLVR rewards throughout the Combination method training iterations. While there is a positive global trend, the outliers indicate that the two reward signals are not always synchronized. These outliers represent conflicting supervision signals, which can cause the combination method to underperform compared to the pure process-level supervision provided by DRM._
>
> ---
>
> We hope that our responses and the additional evidence presented here make the contributions and technical soundness of the work clearer. We have uploaded a revised version of our paper, in which the newly added content is highlighted in blue for convenience. If you have any further questions or would like to discuss anything in more detail, we’d be very happy to continue the conversation.

---

> ### Author Response · Authors · 2025-11-27
> **Following up on our response to your concerns**
>
> Dear Reviewer ubnr,
>
> Thank you for your time and constructive comments.
>
> As the discussion period is ending soon (Dec 3rd), we would like to kindly check if our previous response and the revised paper have addressed your concerns.
>
> In our rebuttal, we have made specific efforts to clarify the points you raised (especially regarding the integration of dimensions, the alternative implementations of Confidence, and the in‑depth analysis of the combination of RLVR and DRM supervision). We hope the new results and explanations demonstrate the soundness and contribution of our work.
>
> We value your feedback and would appreciate it if you could let us know if there are any concerns. We are happy to engage in further discussion.
>
> Best regards,
>
> The Authors

---

### Official Review · Reviewer_JprV · 2025-11-01

**Soundness:** 3
**Presentation:** 3
**Contribution:** 3
**Rating:** 8
**Confidence:** 4

**Summary:**

In this work, the authors propose a novel framework that assesses the quality of the reasoning process along three dimensions: (1) Confidence for uncertainty calibration, (2) Relevance for semantic alignment, and (3) Coherence for logical consistency. Through extensive experiments, the authors show that Dimension-level Reward Model (DRM) can successfully provide supervision signals for both off-policy and on-policy optimization.

**Strengths:**

1. This work innovatively proposes Dimension-level Reward Model for both off-policy and on-policy optimization, and demonstrates the effectiveness of incorporating metrics of reasoning process (e.g., Confidence, Relevance, and Coherence) over vanilla outcome reward.
2. This work has done extensive experiments on the advantage of DRM, revealing new findings on process reward.
3. This work introduces a baseline to merge both process and outcome rewards for on-policy optimization, i.e., simply adding the advantage of both rewards. This opens up a new line of research, and is a significant contribution.

**Weaknesses:**

This work does not investigate deeply how to design a good process metric. Although the dimension-level ones (i.e., Confidence, Relevance, and Coherence) are proposed, more design choices should be compared in calculating the process reward metrics. Also, the final results heavily depend on the accuracy of the process metrics. For example, in cases where the Confidence score mistakenly assigns a flawed reasoning process with a high score, the RL training would be negatively affected. From this perspective, the authors should discuss more on how to calculate the scores.

**Questions:**

See weaknesses.

---

> ### Author Response · Authors · 2025-11-21
> **Rebuttal to Reviewer JprV (Part 1/2)**
>
> (Due to the character limit, we split our response into two parts. This is Part 1.)
>
> ---
>
> We are grateful for your constructive comments and appreciation of our work. Your positive feedback is highly encouraging. We appreciate the time taken to review our manuscript and have addressed the raised points as follows.
>
> ## Question.1
>
> > This work does not investigate deeply how to design a good process metric. Although the dimension-level ones (i.e., Confidence, Relevance, and Coherence) are proposed, more design choices should be compared in calculating the process reward metrics.
>
> We address the question of "how to design a good process metric" by focusing on two aspects: choosing the right evaluation dimensions and figuring out how to combine them.
>
> ### Selection of Dimensions
>
> Specifically, we employ Confidence, Relevance, and Coherence to cover the aspects of uncertainty, semantics and logic. However, a key contribution of our work is that we propose the DRM as a generalizable paradigm rather than a static collection of metrics. Our DRM framework is inherently modular, allowing to conveniently introduce new design choices or domain-specific dimensions. This extensibility ensures that our framework can evolve to incorporate "more design choices" as you kindly suggested. To support this extensibility, we establish a systematic validation pipeline. This pipeline serves to (1) verify whether the dimension can identify superior reasoning processes, (2) validate the effectiveness of the supervision signal, and (3) explore the optimal strategy for its utilization. Our experiments are organized precisely following this paradigm: Evaluating Whether DRM Guides Correct Answers (Section 3.2), Assessing the Effectiveness of DRM Supervision (Section 3.3), and Enhancing RLVR with DRM (Section 3.4). Consequently, any potential dimension can be systematically introduced and validated through this paradigm in our framework.
>
> ### Integration of Dimensions
>
> In our experiments, we conduct a comprehensive ablation study on different aggregation methods in Section3.2. Specifically, we examine the performance of each dimension (Confidence, Relevance, Coherence) independently and compare them against combined methods, including equal weighting and our proposed fixed weighting. We select the final model output based on the highest process reward under each setting and measure answer correctness. The results are shown in Table3. The results demonstrate that combining dimensions consistently outperforms single-dimension metrics, and **the performance remains robust across different weighting schemes and model backbones**.
>
> > Table3 (Page5)
> >
> > Answer correctness (\%) of DRM construction approaches on RewardBench2. Native means the performance of the backbone models. (0.1,0.2,0.7) means weights for Confidence, Relevance and Coherence are 0.1, 0.2, 0.7, respectively. LTR denotes that a Learning-to-Rank model is employed for the construction. The highest result in each row is in **bold**.
> > | Model | Native | Confidence | Relevance | Coherence | Weighted Equally | Weighted (0.1,0.2,0.7) | LTR |
> > | :------------------- | :----- | :--------- | :-------- | :-------- | :--------------- | :--------------------- | :-------- |
> > | LLaMA3.1-8B-Instruct | 67.17 | 65.44 | 72.32 | 78.55 | 77.45 | 78.57 | **79.13** |
> > | R1-Distil-Llama8B | 63.46 | 63.10 | 66.76 | **76.35** | 75.11 | 76.16 | 75.18 |
> > | Qwen3-8B | 84.87 | 83.20 | 85.10 | 85.54 | 85.01 | 85.65 | **85.88** |

---

> ### Author Response · Authors · 2025-11-21
> **Rebuttal to Reviewer JprV (Part 2/2)**
>
> (Due to the character limit, we split our response into two parts. This is Part 2.)
>
> ---
>
> We further verify this robustness on the HotpotQA with RAG dataset to assess **robustness across different data distributions** in Appendix G.1. As shown in Table11, the combined methods maintain consistent effectiveness compared to individual dimensions, reinforcing the stability and robustness of our design.
>
> > Table11 (Page27)
> >
> > Answer correctness (\%) of DRM construction approaches on HotpotQA with RAG. Native means the performance of the backbone models. (0.1,0.2,0.7) means weights for Confidence, Relevance and Coherence are 0.1, 0.2, 0.7, respectively. LTR denotes that a Learning-to-Rank model is employed for the construction. The highest result in each row is in **bold**.
> > | Model | Native | Confidence | Relevance | Coherence | Weighted Equally | Weighted (0.1,0.2,0.7) | LTR |
> > | :--- | :--- | :--- | :--- | :--- | :--- | :--- | :--- |
> > | LLaMA3.1-8B-Instruct | 45.31 | 52.42 | 54.56 | 61.36 | 61.33 | **61.70** | 61.25 |
> > | R1-Distil-Llama8B | 43.09 | 49.77 | 47.90 | 55.58 | 55.49 | 55.58 | **55.76** |
> > | Qwen3-8B | 63.61 | 63.37 | 64.36 | 64.31 | **64.55** | 64.39 | 64.11 |
>
> Inspired by your kind suggestion, we explore moving **beyond fixed linear weights to learnable weighting mechanisms**. In particular, we implement an additional Learning-to-Rank (LTR) approach, treating the dimension-wise scores as features and framing selection as a reranking task. LTR selects the reasoning process ranked at the top. The results of this experiment are listed in the "LTR" column of Table3 and Table11. Together with the linear-weight experiments, we observe that **despite employing different weighting strategies, the results remain consistently similar across backbone models and training datasets**. This consistency further indicates the **robustness of our overall framework**.
>
> For clarity and simplicity, we therefore retain the linear-weight combined methods in the main paper, as we believe a stable and interpretable weighting approach is more "elegant", particularly when learnable methods (LTR) do not yield noticeable gains.
>
> ---
>
> ## Question.2
>
> > Also, the final results heavily depend on the accuracy of the process metrics. For example, in cases where the Confidence score mistakenly assigns a flawed reasoning process with a high score, the RL training would be negatively affected.
>
> In our experiments, we observe that Confidence score alone occasionally assigns high scores to flawed reasoning. However, our three proposed dimensions **capture complementary aspects of the reasoning process and rarely misjudge the same instance simultaneously**. This multi-dimensional design **mitigates the impact of isolated scoring errors**. As Figure3(a) in Section4.1 demonstrates, DRM effectively reduces the occurrence of "correct answers with flawed reasoning". This empirical evidence confirms that the combined process methods can guide the RL training effectively.
>
> ---
>
> Once again, we are very grateful for your positive assessment and constructive comments. We have uploaded a revised version of our paper, in which the newly added content is highlighted in blue for convenience. If you have any further questions or would like to discuss anything in more detail, we’d be very happy to continue the conversation.

---

> ### Author Response · Authors · 2025-11-27
> **Following up on the discussion**
>
> Dear Reviewer JprV,
>
> Thank you again for your positive assessment and strong support for our work.
>
> Following the AC's suggestion, we are writing to check if you have any remaining questions regarding our rebuttal or the revised paper. We have carefully polished the paper based on the feedback, and we are fully prepared to provide further clarifications if needed.
>
> We truly appreciate your time and recognition of our contribution.
>
> Best regards,
>
> The Authors

---

### Comment · Area_Chair_7iMw · 2025-11-26

Dear Reviewers,

Thank you once again for your service to ICLR 2026. Now that the authors have submitted their rebuttal, I kindly ask you to take the following steps (if you have not done so already):

Read the authors’ response and other reviews.
Consider whether the rebuttal and additional comments affect your assessment of the paper.
Engage in interactive discussion with the authors -- Note the Author-Reviewer-AC discussion period ends on 12/3 9 PM UTC. You are recommended to keep active before that deadline. If you have more concerns/questions (e.g., requesting clarifications, new results), it is recommended to post your request asap, so that the authors have enough time to address them.
The current reviews for this paper are mixed (scores: 4/8/4). Your further contributions are essential for forming a well-informed final decision.

I am happy to join and support the discussions between you and the authors. Please feel free to share your thoughts and participate actively in the discussion. Thanks!

Best regards,

The AC

---

### Author Response · Authors · 2025-12-03
**General Response: Summary of Our Rebuttal (Part 1/2)**

(Due to the character limit, we split our general response into two parts. This is Part 1.)

---

We sincerely express our gratitude to all reviewers for their time and constructive feedback. We are encouraged that they find our Dimension-level Reward Model (DRM) **"innovative"** with **"a significant contribution" (Reviewer JprV)** targeting **"known gaps"** in LLM optimization. Reviewers highlight that our framework **"outperforms native and strong baselines" (Reviewer ubnr)** and provides a **"reasonable"** and **"effective" (Reviewer pnbE)** supervision of the reasoning process. **During the discussion period, we have a productive exchange with Reviewer pnbE, who acknowledges our clarifications and agrees to raise the score.** Here we quote:

> As for changing my score, I’m not planning to revise it before ICLR announces its official resolution to this incident. Sorry about that. **But AC could see my score as 5 at least.**

Below, we summarize our responses. The reviewers raise one common concern and four specific concerns. We reply to each point in detail. Our discussion with Reviewer pnbE confirms that both the common concern and the individual ones are resolved. We believe our responses sufficiently address these concerns.

---

## 1. Robustness of DRM Framework (Reviewer JprV, ubnr and pnbE)

_Concerns regarding robustness across weight integration strategies (Reviewer JprV, ubnr and pnbE), model backbones (Reviewer ubnr and pnbE), training data distributions (Reviewer ubnr and pnbE), and tolerance for metric misjudgements (Reviewer JprV)._

Our extensive experiments demonstrate **DRM's robustness across these four aspects.** The results are shown in Table3, Table11 and Figure3(a).

- **Weight Integration Strategies.**

  We compare our fixed weighting strategy with individual dimensions and different weighting methods. The results show that combined dimensions consistently outperform individual ones and remain stable across fixed, equal, and learned (Additional Experiments) weighting schemes, confirming the framework is robust to different integration strategies.

- **Model Backbones.**

  We conduct experiments on three different model backbones (Llama-3.1-8B-Instruct, R1-distil-LLaMA8B and Qwen3-8B). Results show consistent gains across non-reasoning, reasoning, and hybrid-reasoning model backbones, demonstrating the backbone-agnostic effectiveness of our method.

- **Training Data Distributions.**

  We extend our method to HotpotQA with RAG dataset. The results show our method maintains its effectiveness in this new dataset, verifying its generalization to new data distributions.

- **Metric Misjudgements.**

  We emphasize that our multi-dimensional design captures **complementary aspects** of the reasoning process. These dimensions **rarely misjudge the same instance simultaneously**. Empirical analysis confirms the reduction of "correct answers with flawed reasoning", preventing the framework from being driven by any single dimension’s misjudgements.

**Reviewer pnbE confirms this common concern is fully resolved.** As Reviewer pnbE states:

> I really appreciate the additional experiments you added to further demonstrate the effectiveness of your work...The remaining two points you clarified did resolve my concerns.

**Since Reviewer JprV and ubnr share similar concerns regarding robustness, we believe the extensive evidence provided above effectively addresses their questions as well.**

---

> ### Author Response · Authors · 2025-12-03
> **General Response: Summary of Our Rebuttal (Part 2/2)**
>
> (Due to the character limit, we split our general response into two parts. This is Part 2.)
>
> ---
>
> ## 2. Confidence Score Implementations (Reviewer ubnr)
>
> _Concerns regarding alternative log-prob-based confidence score implementations._
>
> We compare our method with two alternative log-prob-based confidence score implementations: **Perplexity** and **Average Token Entropy**. Results show that these alternatives do not outperform our approach (see Table7 and Table8). This indicates that our original design is both effective and efficient.
>
> ---
>
> ## 3. Interaction Effects of RLVR and DRM in GRPO (Reviewer ubnr)
>
> _Concerns regarding the slight performance drop of the RLVR-DRM combination in some tasks._
>
> We examine the training dynamics of the combination approach by **correlating their rewards during training** in Figure 2, which reveals instances where outcome rewards (RLVR) and process rewards (DRM) conflict. This finding indicates that during combination training, the two reward signals sometimes push the model in opposite directions, which explains the slight performance drop in the combination approach. (Additional Experiments)
>
> ---
>
> ## 4. Computational Overhead (Reviewer pnbE)
>
> _Concerns regarding training and inference costs._
>
> - **Inference.**
>
>   DRM is a training-time framework with **zero overhead during inference**.
>
> - **Training.**
>
>   We detail the cost analysis in Appendix E. To reduce costs, we propose DRM-Light, using a lightweight Coherence evaluator. Table 10 shows that DRM-Light maintains competitive performance with **25% more GPU resources and 9% more training time, resulting in only 36% increase in total GPU-hours.** As requested by Reviewer pnbE in our discussion, we further clarify how the 25% figure is computed: adding two evaluator nodes (16 GPUs) to the 64‑GPU RLVR baseline increases total GPU count from 64 to 80, which is exactly a 25% increase in resources.
>
> ---
>
> ## 5. Qualitative Impact on Reasoning (Reviewer pnbE)
>
> _Concerns regarding DRM's impact on the quality of reasoning process._
>
> - **Logical Soundness.**
>
>   We evaluate the logical soundness of model outputs using GPT-4o. Results shown in Figure3(a) demonstrate DRM significantly reduces "correct answers via flawed reasoning" compared to RLVR.
>
> - **Case Study.**
>
>   We conduct a case study comparing samples from RLVR and DRM in Appendix C. The comparison confirms DRM promotes sound logic, whereas RLVR often relies on shortcuts.
>
> - **Reasoning Structure and Efficiency.**
>
>   We perform pairwise judgments on the structure and efficiency of reasoning process against backbone models. Results shown in Figure3(b) indicate that DRM induces a more structured reasoning pattern with only a modest length increase (+4.9%), which is more efficient than RLVR (+12.1%). (Additional Experiments)
>
> **Reviewer pnbE confirms this concern is fully resolved.**
>
> ---
>
> We incorporate all new experiments and analyses into the revised manuscript, where modifications are highlighted in blue for convenience. We are confident that these additions address the reviewers' concerns. Once again, we sincerely thank the Area Chair and all reviewers for their time and constructive feedback. Their comments help us clarify the methodology and strengthen the solidity of our work.

---

### Meta-Review · Area_Chair_y5vv · 2025-12-18

**Summary:**

Reviewer JprV criticized the lack of comprehensive comparisons with other process reward metrics and questioned the reliability of the proposed metrics. Reviewer ubnr raised concerns regarding: 1) robustness to judge/model selection and weight calibration, 2) limited analysis of per-token confidence calibration, and 3) the slight performance degradation observed in combination-supervised training. Reviewer pnbE questioned the significance of the improvements, hyperparameter sensitivity (e.g., reward weights), and the computational overhead of external evaluators.

**Reviewer Concerns:**

The authors try to address the concerns raised by Reviewer JprV by providing additional experiment results to validate the robustness of the proposed method across different weighted strategy, LLMs and domains. However, the reviewer's concerns about lack of comparsion with more design choices in calculating the process reward metrics seems still remain.

For reviewer ubnr, the authors have provided extensive experiments to validate the robustness of the proposed methods on different weighting schemes and model backbones. Besides, the authors also conducted additional experiments considering more confidence estimator according to the reviewers' suggestion and clearly explain the reasons of the performance drop, AC think the new results strengthen the manusctipt. However, since the modification made during the rebuttal is rather extensive, AC is not sure about whether the original concerns can be well-addressed and integrated into the final revision.

For reviewer pnbE, given the additional experiments, the reviewer's concern about the sensitiveness of hyperparameters can be well addressed. However, AC think the reviewer's concern about the training cost and dependency on serveral external evaluators still exists after the discussion.

**Reviewer Scores:**

Reviewer JprV's concerns are minor. The authors provide very comprehensive response. And thus he/she may maintain his/her original rating after discussion.

Reviewer ubnr and pnbE provided constructive and in-depth review. The authors made great efforts in addressing his/her concerns. The AC appreciates the heavy workload managed by the authors during the review period and acknowledges that these results help strengthen the reliability of the paper. From AC's own reading from this manuscript, the proposed method is promising and inspiring. However, given the substantial volume of new experimental results and the fact that 1) the required revisions are not minor and 2) some concerns seem still exist after discussion, AC is not sure that the reviewers will change their scores to positive. And thus this paper may not be ready to be published on ICLR in its current form. The authors are encouraged to revise this work according the the reviewers' suggestions and resubmit to other top-venues.

---

### Decision · Program_Chairs · 2026-01-26

Reject